# SBGD: Improving Graph Diffusion Generative Model via Stochastic Block Diffusion

Junwei Su [1]   Shan Wu [2]

## Abstract

Graph diffusion generative models (GDGMs) have emerged as powerful tools for generating high-quality graphs. However, their broader adoption faces challenges in *scalability and size generalization*. GDGMs struggle to scale to large graphs due to their high memory requirements, as they typically operate in the full graph space, requiring the entire graph to be stored in memory during training and inference. This constraint limits their feasibility for large-scale real-world graphs. GDGMs also exhibit poor size generalization, with limited ability to generate graphs of sizes different from those in the training data, restricting their adaptability across diverse applications. To address these challenges, we propose the stochastic block graph diffusion (SBGD) model, which refines graph representations into a block graph space. This space incorporates structural priors based on real-world graph patterns, significantly reducing memory complexity and enabling scalability to large graphs. The block representation also improves size generalization by capturing fundamental graph structures. Empirical results show that SBGD achieves significant memory improvements (up to $6\times$) while maintaining comparable or even superior graph generation performance relative to state-of-the-art methods. Furthermore, experiments demonstrate that SBGD better generalizes to unseen graph sizes. The significance of SBGD extends beyond being a scalable and effective GDGM; it also exemplifies the principle of modularization in generative modeling, offering a new avenue for exploring generative models by decomposing complex tasks into more manageable components.

[1]School of Computing and Data Science, University of Hong Kong [2]School of Resources and Environmental Engineering, Hefei University of Technology. Correspondence to: Junwei Su <junweisu@connect.hku.hk>, Shan Wu <wus@hfut.edu.cn >.

*Proceedings of the $42^{nd}$ International Conference on Machine Learning*, Vancouver, Canada. PMLR 267, 2025. Copyright 2025 by the author(s).

## 1. Introduction

Graphs are a fundamental mathematical construct used to represent relational data, consisting of nodes and edges that depict pairwise relationships. This concept is widely applied across various fields, such as social networks (Grover et al., 2019; Wang et al., 2018), program synthesis (Brockschmidt et al., 2018), and neural architecture search (Xie et al., 2019; Lee et al., 2021). A key challenge in this domain is the generation of new graphs that mirror the properties of the observed ones. For example, in drug discovery, this involves creating graphs that represent the structural composition of specific proteins(Simonovsky & Komodakis, 2018; Li et al., 2018; Preuer et al., 2018) or molecules.

Given its significance, the graph generation problem has a long history of research, with rule-based random graph models traditionally dominating the field (Barabási & Albert, 1999; Holland et al., 1983; Erdős et al., 1960; Newman et al., 2002). *A prime example of such a model is the Stochastic Block Model (SBM)(Holland et al., 1983), which is based on the observation that real-life graphs often consist of densely connected blocks of vertices exhibiting similar behaviors*(Abbe, 2018; Newman et al., 2002; Newman & Girvan, 2004; Newman, 2006; Karrer & Newman, 2011; Cherifi et al., 2019; Su & Marbach, 2022). This block structure reflects community-like patterns where nodes within the same block are more likely to connect, while inter-block connections are sparser. However, these rule-based models face significant limitations: they fail to capture the complex and nuanced distribution of graph-structured data observed in real-world problems (Niu et al., 2020). As a result, the focus has shifted towards deep learning-based methods that can model more intricate graph properties. Among these, graph diffusion generative models (GDGMs) have emerged as a promising solution, showing impressive performance in graph generation (Niu et al., 2020; Vignac et al., 2022; Jo et al., 2022).

GDGMs belong to the family of diffusion-based generative models (also referred to as the score-based generative models), which model the data generation process through a diffusion mechanism operating within the data space. These models consist of two key components: (1) a forward process that gradually degrades the data into a simple known

distribution, such as standard Gaussian, and (2) a neural network that reverses this process to reconstruct the original data (see Sec. 2 for more details). Diffusion models are known for their ability to generate high-quality samples and have been successfully applied in domains like image and audio synthesis (Dhariwal & Nichol, 2021; Rombach et al., 2022; Nichol et al., 2021; Yang et al., 2023). Extending this framework to graph data has produced promising results, as these models are capable of capturing the complex distributions and structures characteristic of graphs.

**Challenges and Limitations.** Despite their success, existing GDGMs face two primary challenges: *scalability and size generalization.* First, scalability remains a significant hurdle due to the high memory demands of GDGMs. These models currently model the diffusion generative process at the graph level, requiring the entire graph — including its structure and features — to be stored in memory during the process. As the graph size increases, memory usage scales quadratically with the number of nodes, making these models impractical for large-scale graphs, such as those found in social networks or molecular simulations, where graphs can contain millions of nodes and edges (Hamilton et al., 2018). Second, GDGMs face difficulties with size generalization. They are typically trained on graphs of a fixed size and often fail to generate graphs that differ significantly in size from those encountered during training. This limitation restricts their applicability in scenarios where graph sizes can vary widely, such as in molecular structures (Wieder et al., 2020).

### 1.1. Contribution and Significance

To address the challenges outlined, we propose the Stochastic Block Graph Diffusion (SBGD) model, which incorporates a structural prior based on the block-based organization commonly observed in real-world graphs (see Figure 4 for an illustration). By leveraging this block structure, SBGD captures community-like patterns and represents the graph in a block space. Performing the diffusion process within this block space significantly reduces memory usage, enabling the generation of larger graphs with reduced memory overhead (see Table 1 for a comparison of memory complexity between our method and existing GDGMs). Additionally, this block-based representation enhances the model's ability to generalize across graphs of varying sizes, as smaller, fundamental building blocks can be recombined in different configurations to generate graphs at different scales.

Empirical evaluations on both real-world and synthetic datasets show that SBGD achieves up to a $6\times$ improvement in memory efficiency, while maintaining comparable or superior generative performance relative to state-of-the-art GDGMs. This makes SBGD highly scalable for large graphs. Furthermore, experiments on size generalization demonstrate that SBGD exhibit better size generation, par-

ticularly exceling at generating graphs larger than those seen in the training set. Our ablation study on block graph size also reveals that smaller block representations initially improve performance, but excessively small blocks can degrade generative quality. Moreover, experiments show that block graph size also impacts the model's size generalization ability. These findings suggest the existence of an optimal block size, with granularity depending on the data's properties and the specific generative task. This insight opens up promising avenues for future research.

Overall, these results highlight the advantages of incorporating a structural prior through block representation, allowing SBGD to overcome key limitations of existing diffusion-based models. This approach provides a more flexible, scalable, and efficient solution to graph generation. Furthermore, *it exemplifies the principle of modularization in generative modeling, offering a novel way to explore generative models by decomposing complex tasks into more manageable components.*

## 2. Background and Motivation

In this section, we present a brief introduction to diffusion-based generative models, graph diffusion generative models, and the role of block structure in graphs. We defer some of the technical details to the appendix due to page limitation.

### 2.1. Diffusion Generative Models.

DGMs (Song & Ermon, 2019; Song et al., 2020b; Song & Ermon, 2020; Sohl-Dickstein et al., 2015; Song et al., 2020a) belong to the large family of latent variable models. It models the data generation process as a transition process in the latent space and primarily comprises two main components: a noise model (forward process) and a denoising neural network (backward process). The noise model $\mathbb{P}$ gradually corrupts a data point $\mathbf{x}$ to form a sequence of increasingly noisy data points $(\mathbf{x}_1, \ldots, \mathbf{x}_T)$. It adheres to a Markovian structure, formulated as:

$$\mathbb{P}(\mathbf{x}_1, \ldots, \mathbf{x}_T | \mathbf{x}) = \mathbb{P}(\mathbf{x}_1 | \mathbf{x}) \prod_{t=2}^{T} \mathbb{P}(\mathbf{x}_t | \mathbf{x}_{t-1}).$$

In most cases, Gaussian noise is used for the forward process. This amounts to a Markov chain that gradually adds Gaussian noise to the data according to a variance schedule $\beta_1, \ldots, \beta_T$:

$$\mathbb{P}(\mathbf{x}_t | \mathbf{x}_{t-1}) := \mathbb{N}(\mathbf{x}_t; \sqrt{1 - \beta_t} \mathbf{x}_{t-1}, \beta_t \mathbf{I}),$$
$$\mathbb{P}(\mathbf{x}_t | \mathbf{x}_0) = \mathbb{N}(\mathbf{x}_t; \bar{\alpha}_t \mathbf{x}_0, (1 - \bar{\alpha}_t) \mathbf{I}),$$

where $\alpha_t := 1 - \beta_t$ and $\bar{\alpha}_t := \prod_{s=1}^{t} \alpha_s$. Then, the goal of the backward process is to learn a (denoising) neural network $s_{\boldsymbol{\theta}}$ ($\boldsymbol{\theta}$ is the learnable parameter) to reverse this noising process.

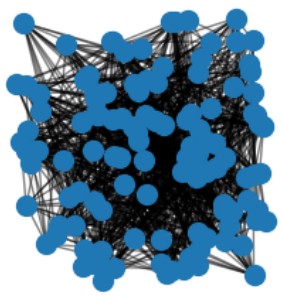

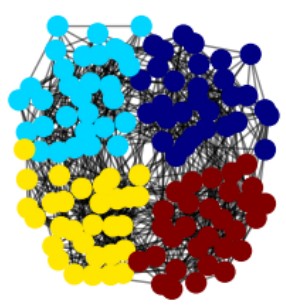

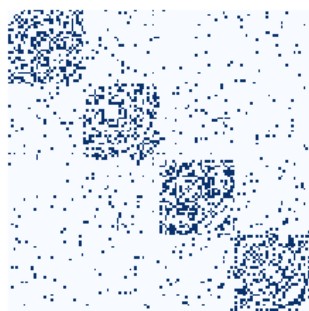

(a) Example graph with random layout.

(b) Example graph with block structure highlighted.

(c) Adjacency matrix of the example graph.

*Figure 1.* Visualization of an example graph in different layouts and its matrix representation. Fig. 1(a) is a visualization of the example graph with a random layout. Fig. 1(b) is a visualization of the example graph with a Kamada Kawai layout (Kamada et al., 1989) and the blocks are highlighted with different colours. Fig. 1(b) is a visualization of the adjacent matrix of the example graph where the dense area in the diagonal region of the matrix represent the blocks in the graph.

| Graph Diffusion Generative Methods | SBGM (Niu et al., 2020) | SDE (Jo et al., 2022) | DiGress (Vignac et al., 2022) | SBGD (ours) |
|---|---|---|---|---|
| Memory Complexity | $\mathcal{O}(N^2)$ | $\mathcal{O}(N^2 + NF)$ | $\mathcal{O}(N^2 + NF)$ | $\mathcal{O}(C^2 + CF)$ |
| Computation Complexity | $\mathcal{O}(TN^2)$ | $\mathcal{O}(T(N^2 + NF))$ | $\mathcal{O}(T(N^2 + NF))$ | $\mathcal{O}(T(kC^2 + kCF))$ |

*Table 1.* A summarized comparison of existing graph diffusion generative model. $N$ is the number of vertices and $F$ is the dimension of the feature. $C$ is the size of the block graph (which is much smaller than $N$) and $k$ is the number of blocks from the graph ($kC$ is the size of the graph). $T$ is the number of diffusion steps used in the generation process. It is evident from the table that our method offers better memory complexity while keeping the same computation complexity.

To generate new samples, a point is sampled from the prior (convergent) distribution $\mathbb{P}(\mathbf{x}_T)$ (typically standard Gaussian) and iteratively denoised by the neural network $s_{\boldsymbol{\theta}}$ until it recovers the original data distribution. This follows a series of (reverse) state transitions,

$$\mathbf{x}_T \xrightarrow{s_{\boldsymbol{\theta}}} \mathbf{x}_{T-1} \xrightarrow{s_{\boldsymbol{\theta}}} \cdots \xrightarrow{s_{\boldsymbol{\theta}}} \mathbf{x}_0.$$

This is achieved by using sampling rules such as those specified in DDPM/DDIM (Ho et al., 2020; Song et al., 2020a).

GDGMs represent an important extension of DDMs to graph data and have demonstrated empirical success in graph generation tasks. Due to the inherent complexity of graph data, which consists of both structural and feature information, existing GDGM extensions rely on matrix representations of the graph. They model the graph generation process using a system of stochastic processes that capture the dependency between the structure and features. This is typically achieved by incorporating underlying models, such as graph neural networks, to model these interdependencies. The stochastic process used in GDGMs can either be a stochastic differential equation (for continuous data spaces) (Jo et al., 2022) or a discrete Markov chain (for discrete data spaces) (Vignac et al., 2022). Both approaches model the graph generation by considering the entire graph at each step, encompassing both the structure and feature matrix.

### 2.2. Block Structure in Graphs.

In real-life graphs, a prominent feature is their block (community structure) (Deshpande et al., 2018; Holland et al., 1983), where vertices within the same block display dense connections and exhibit similar behavioural traits (i.e., similar feature distributions). This contrasts sharply with vertices from different blocks, which are sparsely connected, highlighting the distinct boundaries between these units. For example, in social networks, groups of friends form tightly connected communities. In citation networks, papers in the same research domain cite each other frequently. In biological networks, where certain proteins or genes interact heavily within specific functional modules. Block structures are a cornerstone of graph analysis (Newman et al., 2002; Newman, 2006) and play a critical role in many graph learning algorithms, such as DeepWalk (Perozzi et al., 2014) and ClusterGCN (Chiang et al., 2019). The block structure simplifies graph representation, highlighting modularity and reducing complexity by focusing on inter- and intra-block interactions.

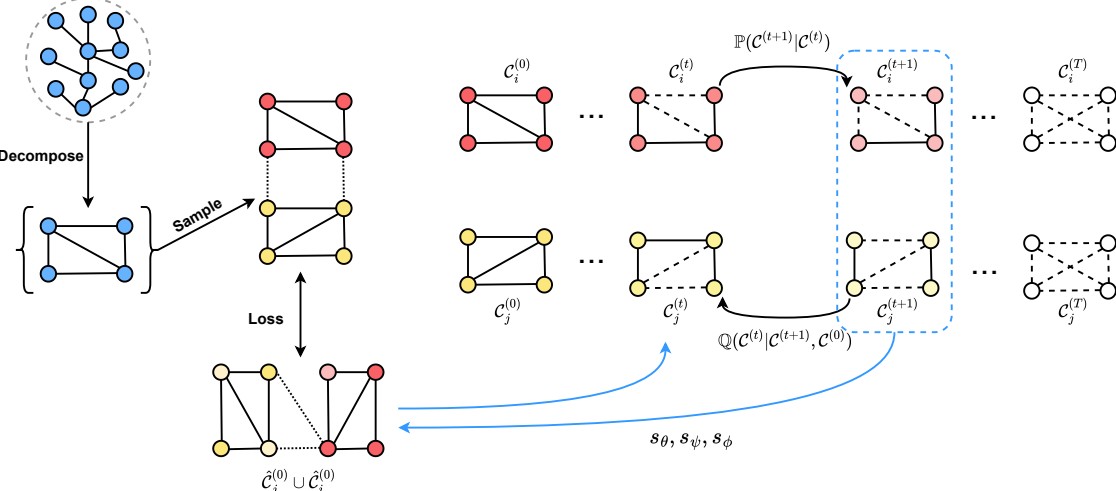

*Figure 2.* The overview of SBGD. (1) incoming graphs are decomposed into block graphs where vertices share similar properties. (2) sample from the set of block graphs and conduct separate diffusion. The noise model is defined by the distribution $\mathbb{P}(.)$ (3) The denoising network $s_{\boldsymbol{\theta}}, s_{\psi}, s_{\phi}$ learns to predict the clean graph, including the inter-connections, from $\mathcal{C}_j^{(t)}$, and $\mathcal{C}_i^{(t)}$. During inference, the predicted distribution is combined with $\mathbb{Q}(\mathcal{C}^{(t-1)}|\mathcal{G}, \mathcal{G}^{(t)})$ in order to sample a discrete $\mathcal{C}^{(t-1)}$ from this distributions.

## 3. Stochastic Block Graph Diffusion (SBGD)

In this section, we introduce the SBGD model, designed to overcome the scalability and size generalization challenges of traditional GDGMs by incorporating block structure principles into the diffusion-based graph generation framework. SBGD performs the diffusion process within a compact block space instead of the full graph space, thereby reducing memory complexity and enhancing size generalization. Fig. 2 is an overview of SBGD.

### 3.1. Block Graphs Representation

Let $\mathcal{G} = (\mathcal{V}, \mathcal{E}, \mathbf{X})$ denote an undirected graph, where $\mathcal{V} = \{1, 2, \ldots, N\}$ is the set of $N$ nodes, $\mathcal{E} \subseteq \mathcal{V} \times \mathcal{V}$ represents the edges, and $\mathbf{X} \in \mathbb{R}^{N \times F}$ is the node attribute matrix, where each row $\mathbf{x}_v$ contains the feature vector for node $v$. Alternatively, we can represent $\mathcal{G}$ as a two-tuple $\mathcal{G} = (\mathbf{A}, \mathbf{X})$, where $\mathbf{A}$ is the $N \times N$ adjacency matrix, with $\mathbf{A}[i, j] = 1$ if there is an edge between nodes $i$ and $j$, and $0$ otherwise. These representations are used interchangeably to accommodate different aspects of the generation process. For any matrix $\mathbf{H}$, we denote its transpose as $\mathbf{H}'$ and its value at time step $t$ as $\mathbf{H}^{(t)}$.

To apply block decomposition, we partition the nodes into $k$ non-overlapping groups, $\mathcal{V} = [\mathcal{V}_1, \ldots, \mathcal{V}_k]$, where $\mathcal{V}_i$ contains the nodes in the $i$-th partition (see the supplementary material for partition details). The subgraph induced by $\mathcal{V}_i$ is denoted as the block graph $\mathcal{C}_i = \mathcal{G}[\mathcal{V}_i]$. Given this partition, we have a set $\mathcal{C}$ of $k$ block graphs as

$$
\begin{aligned}
\mathcal{C} &= [\mathcal{C}_1, \ldots, \mathcal{C}_k] \\
&= [\mathcal{G}[\mathcal{V}_1], \ldots, \mathcal{G}[\mathcal{V}_k]] \\
&= [(\mathbf{A}_1, \mathbf{X}_1), \ldots, (\mathbf{A}_k, \mathbf{X}_k)],
\end{aligned}
$$

where $\mathbf{A}_i$ and $\mathbf{X}_i$ are the adjacency matrix and feature matrix of block graph $\mathcal{C}_i$, respectively. The complete adjacency matrix $\mathbf{A}$ of $\mathcal{G}$ can be partitioned into $k^2$ submatrices:

$$
\mathbf{A} = \bar{\mathbf{A}} + \boldsymbol{\Delta} = \begin{bmatrix} \mathbf{A}_1 & \cdots & \mathbf{A}_{1k} \\ \vdots & \ddots & \vdots \\ \mathbf{A}_{k_1} & \cdots & \mathbf{A}_k \end{bmatrix}
$$

$$
\bar{\mathbf{A}} = \begin{bmatrix} \mathbf{A}_1 & \cdots & 0 \\ \vdots & \ddots & \vdots \\ 0 & \cdots & \mathbf{A}_k \end{bmatrix}, \quad \boldsymbol{\Delta} = \begin{bmatrix} 0 & \cdots & \mathbf{A}_{1k} \\ \vdots & \ddots & \vdots \\ \mathbf{A}_{k1} & \cdots & 0 \end{bmatrix}.
$$

Here, $\bar{\mathbf{A}}$ consisting of all diagonal blocks of $\mathbf{A}$ and each diagonal block $\mathbf{A}_i$ is a $|\mathcal{V}_i| \times |\mathcal{V}_i|$ adjacency matrix containing the links within $\mathcal{C}_i$. $\boldsymbol{\Delta}$ is the matrix consisting of all off-diagonal blocks of $\mathbf{A}$ and $\mathbf{A}_{ij}$ contains the links between two partitions $\mathcal{V}_i$ and $\mathcal{V}_j$. Then, the graph $\mathcal{G}$ can be represented by two tuples $(\{\mathcal{C}_i\}, \{\mathbf{A}_{ij}\})$ where $\{\mathcal{C}_i\}$ is the set of block graphs and $\{\mathbf{A}_{ij}\}$ is set of adjacency matrix among block graphs with the sparse connection.

### 3.2. Diffusion Frameworks

**Forward process.** Based on the block graph decomposition, we further factorize the distribution of feature matrix

and adjacency for the forward process as:

$$\mathbb{P}(\mathcal{G}^{(t)}|\mathcal{G}^{(t-1)}) = \prod_{i \in [k]} \mathbb{P}(\mathcal{C}_i^{(t)}|\mathcal{C}_i^{(t-1)}) \prod_{i,j \in [k]} \mathbb{P}(\mathbf{A}_{ij}^{(t)}|\mathbf{A}_{ij}^{(t-1)}).$$
(3.1)

The complete distribution of the graph is decomposed into two components where the first component is the distribution of the block graph and the second component is the distribution of the interactions among the block graphs. For the first component, we further decompose the block graph distribution into adjacency matrix and feature matrix, i.e,

$$\mathbb{P}(\mathcal{C}_i^{(t)}|\mathcal{C}_i^{(t-1)}) = \mathbb{P}(\mathbf{X}_i^{(t)}|\mathbf{X}_i^{(t-1)})\mathbb{P}(\mathbf{A}_i^{(t)}|\mathbf{A}_i^{(t-1)}).$$

Then, we adopt the existing approaches from (Jo et al., 2022; Vignac et al., 2022) to model the forward diffusion as a system of stochastic process. We apply Gaussian noise to $\mathbf{X}$ and $\mathbf{A}$ in the forward process, i.e.,

$$\mathbb{P}(\mathbf{A}_i^{(t)}|\mathbf{A}_i^{(t-1)}) = \mathbb{N}(\mathbf{A}_i^{(t)}; \sqrt{(1-\beta_t)}\mathbf{A}_i^{(t-1)}, \beta_t \mathbf{I})$$
$$\mathbb{P}(\mathbf{X}_i^{(t)}|\mathbf{X}_i^{(t-1)}) = \mathbb{N}(\mathbf{X}_i^{(t)}; \sqrt{(1-\beta_t)}\mathbf{X}_i^{(t-1)}, \beta_t \mathbf{I})$$

For the second component of Eq. 3.1, we note that the interactions $\mathbf{A}_{ij}$ between the block graphs are sparse (see Fig. 4), and a light-weight module should be sufficient to capture this information. In addition, the interconnection $\mathbf{A}_{ij}$ depends on the choice of block graphs. To capture such a dependency, we propose to model these interactions matrix with another neural network that takes in two (generated) block graphs $\mathcal{C}_i, \mathcal{C}_j$ as input and generated the interaction among block graph $\mathcal{C}_i$ and $\mathcal{C}_j$ as output.

**Training Objective.** The core idea of the denoising diffusion model is to use neural networks to model the reverse (denoising) process for facilitating generation. With the decomposition above, we need two separate denoising neural networks $s_{\boldsymbol{\theta}}$ and $s_{\boldsymbol{\psi}}$ ($\boldsymbol{\theta}$ and $\boldsymbol{\psi}$ are learnable parameter) to model the denoising processes for the graph structure and feature correspondingly. Instead of directly learning a neural net to model the transition from $t$ to $t-1$, we adopt the idea from (Vignac et al., 2022; Chen et al., 2022) and learn a neural network to predict $\mathcal{C}_i^{(0)}$ (or the noise $\epsilon$) from $\mathcal{C}_i^{(t)}$. This amounts to,

$$\mathcal{L}_{\mathbf{A}_i} = \mathbb{E}_{t,\epsilon}\left[\left\|s_{\boldsymbol{\theta}}(\mathbf{A}_i^{(t)}, \mathbf{X}_i^{(t)}, t) - \mathbf{A}_i^{(0)}\right\|^2\right],$$

$$\mathcal{L}_{\mathbf{X}_i} = \mathbb{E}_{t,\epsilon}\left[\left\|s_{\boldsymbol{\psi}}(\mathbf{A}_i^{(t)}, \mathbf{X}_i^{(t)}, t) - \mathbf{X}_i^{(0)}\right\|^2\right],$$

where $\epsilon$ is drawn from the standard Gaussian distribution. If the denoising neural network simply operates separately in the individual block graph, the sparse connection among the block graphs will be lost. Therefore, in capturing the interconnection, the denoising neural network takes in two

block graphs $\mathcal{C}_i, \mathcal{C}_j$ from the block graph set and aims to predict the clean two-block graph $\mathcal{G}[\mathcal{C}_i] \bigcup \mathcal{G}[\mathcal{C}_j]$ induced by $\mathcal{C}_i, \mathcal{C}_j$. Then, we recover/model the sparse interaction between these two block graphs with another neural $s_{\boldsymbol{\phi}}(\mathcal{C}_i, \mathcal{C}_j)$. This amounts to the following objective,

$$\mathcal{L}_{\mathbf{I}} = \mathbb{E}_{\mathcal{C}_i, \mathcal{C}_j}\left[\|s_{\boldsymbol{\phi}}(\mathcal{C}_i, \mathcal{C}_j) - \mathbf{A}_{ij}\|^2\right].$$

The complete training objective is given by,

$$\mathcal{L} = \mathcal{L}_{\mathbf{A}_i} + \mathcal{L}_{\mathbf{A}_j} + \mathcal{L}_{\mathbf{X}_i} + \mathcal{L}_{\mathbf{X}_j} + \mathcal{L}_{\mathbf{I}}.$$

**Sampling and Implementation.** Once the network $s_{\boldsymbol{\psi}}, s_{\boldsymbol{\theta}}, s_{\boldsymbol{\phi}}$ for the graph structure, features, and cross-block graph interactions are trained, we follow the standard procedure for sampling from a diffusion model. This involves first creating a set of block graphs using $s_{\boldsymbol{\psi}}$ and $s_{\boldsymbol{\theta}}$, and then recovering their interactions using $s_{\boldsymbol{\phi}}$. For our implementation, we use the commonly employed graph transformer architecture for all of these networks. Pseudo-code for training and sampling is provided in Appendix B, along with additional technical details of the implementation.

### 3.3. Theoretical Discussion.

In this section, we present a theoretical discussion of our proposed method. Our analysis focus on the memory complexity and performance benefit of block representation, and the benefit of using analogue bit.

**Memory Analysis.** Existing denoising diffusion-based approaches for graph generation operate directly on the complete graph space, which leads to a memory complexity of $\mathcal{O}(N^2)$, where $N$ is the number of vertices in the graph. This quadratic complexity arises because the diffusion process needs to track pairwise interactions or correlations between every pair of vertices, making it highly memory-intensive for large-scale graphs. In contrast, our proposed SBGD approach reduces memory requirements by decomposing the graph into smaller block graphs, each of size $C$, where $C \ll N$. As a result, the memory complexity is reduced to $\mathcal{O}(C^2)$, which can lead to significant memory savings, especially when dealing with very large graphs. A memory complexity comparison among existing methods is provided in Table 1.

**Advantage for Distributed Training.** This block-wise representation not only reduces the memory footprint but also provides several advantages in the context of distributed training. By partitioning the graph into smaller, subgraphs (blocks), the model can process each pair of block graphs separately or in parallel, allowing for more efficient distributed computation. Each computational node or device in a distributed system can handle smaller subproblems, reducing the memory load per node and mitigating bottlenecks associated with memory constraints.

Therefore, the modularity of the block representation is particularly beneficial for large-scale graph learning tasks. It improves scalability, as the approach scales linearly with the number of blocks rather than quadratically with the number of vertices, making it feasible to handle graphs with millions or even billions of vertices. Moreover, the localized nature of block-wise operations preserves important structural information within each block while enabling inter-block interactions to be modeled through additional aggregation steps, ensuring that the diffusion process still captures global graph properties without incurring prohibitive memory costs.

**Benefit of Structural Prior.** Block representation improves size generalization in graph generation by focusing on local, size-invariant structures like communities, which have consistent internal patterns (e.g., degree distributions, clustering coefficients) regardless of the overall graph size. By learning these modular, block-based components, the model can generalize well across different graph sizes, applying the same learned rules to both smaller and larger graphs. This approach reduces the learning complexity compared to full-graph methods, as it shifts the focus to intra-community relationships (with complexity $\mathcal{O}(C^2)$) and sparse inter-community connections, rather than handling the full graph (with complexity $\mathcal{O}(N^2)$).

Moreover, block-based models avoid overfitting to specific global graph patterns that are unique to certain sizes. This modular approach allows the model to expand or contract based on the number of blocks, making it adaptable to a wide range of graph scales. The combination of reduced learning complexity, modularity, and the ability to learn size-invariant patterns ensures better generalization and faster convergence across graphs of different sizes, enhancing scalability and performance.

## 4. Experiment

In this section, we present an experimental study of our proposed method. We defer some of the technical details of these experiments to the appendix. The goal of the experimental study is to empirically validate and answer the following two main questions for our method.

1. Can SBGD achieve comparable/better performance in generating graphs while requiring less memory?

2. Can SBGD extrapolate better in generating graphs of size that are not observed in training data?

**Overview.** The empirical results answer the above questions affirmly, and thereby validate the effectiveness of SBGD.

### 4.1. General Setup

**Baselines.** In our experiments, we compare the performance of SBGD against several state-of-the-art denoising-diffusion-based graph generation methods including Di-Gress (Vignac et al., 2022), GDSS (Jo et al., 2022), and EDP-GNN (Niu et al., 2020). In addition, we also consider several representative deep graph generation such as GraphRNN (You et al., 2018), SPECTRE (Martinkus et al., 2022), and EDGE (Chen et al., 2023).

**Datasets.** We consider five real and synthetic datasets with varying sizes and connectivity levels: Planar-graphs, Contextual Stochastic Block Model(cSBM) (Deshpande et al., 2018), Proteins (Dobson & Doig, 2003), QM9 (Wu et al., 2018), OGBN-Arxiv, and OGBN-Products (Hu et al., 2021). Notably, OGBN-Products is a large dataset for graph generation tasks, and our method is the only approach capable of successfully training on it.

**Evaluation.** Our evaluation focuses on two main aspects: (1) the quality of the generated graphs, assessing how well the method captures the underlying graph distribution, and (2) the memory consumption required during graph generation. For assessing the quality of generated graphs, we follow established graph generation studies and adopt both structure-based and neural-based metrics. In terms of structure-based metrics, we measure the Maximum Mean Discrepancy (MMD) (Gretton et al., 2012) between test and generated graphs across graph properties, including degrees, clustering coefficients, and orbit counts. For neural-based metrics, we use the Fréchet Inception Distance (FID) proposed by (Heusel et al., 2017) (with modified procedure in (Thompson et al., 2022)) to evaluate the alignment between graph distributions in a learned embedding space. To ensure a direct comparison of memory consumption, we use our method as a reference point and report the memory consumption ratio as the baseline model's memory divided by our model's memory, highlighting the relative efficiency across approaches.

**Testbed.** Our experiments were conducted on a Dell PowerEdge C4140, The key specifications of this server, pertinent to our research, include: **CPU:** Intel Xeon Gold 6230 processors equipped with 20 cores and 40 threads, **GPU:** NVIDIA Tesla V100 SXM2 units equipped with 32GB of memory, **Memory:** An aggregate of 256GB RAM, distributed across eight 32GB RDIMM modules, and **Operating System:** Ubuntu 18.04LTS

### 4.2. Experimental Results

**Effectiveness.** We evaluate the effectiveness of our approach in both real-life and synthetic datasets. The results are presented in Table 1. We observe that SBGD can achieve a significantly better memory consumption (up to $6 \times$ more memory efficient) while maintaining comparable or better performance in generation in almost all metrics compared to existing baselines. In particular, under the given testbed, our method is the only one that can successfully

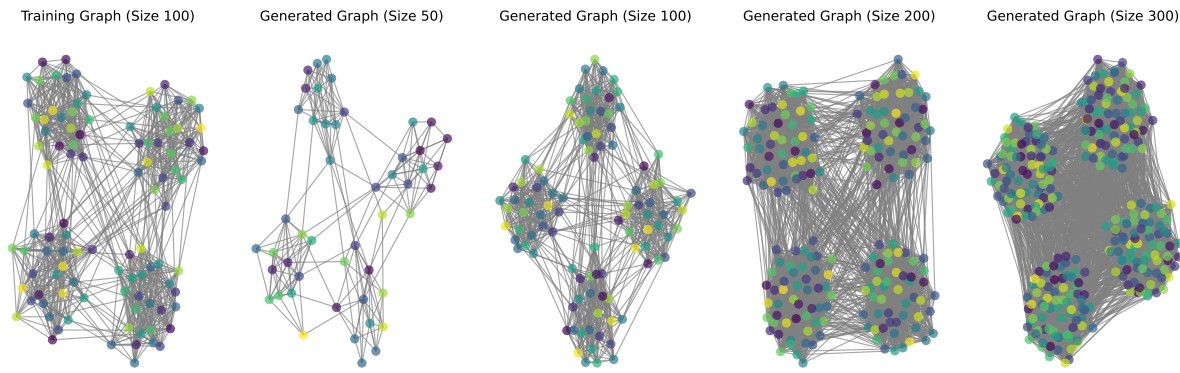

*Figure 3.* Visualization of graphs of different sizes generated from SBGD. The figure illustrates that SBGD is able to maintain the overall characteristic of the ground-truth (training graph) nicely even if generating graph of varied size.

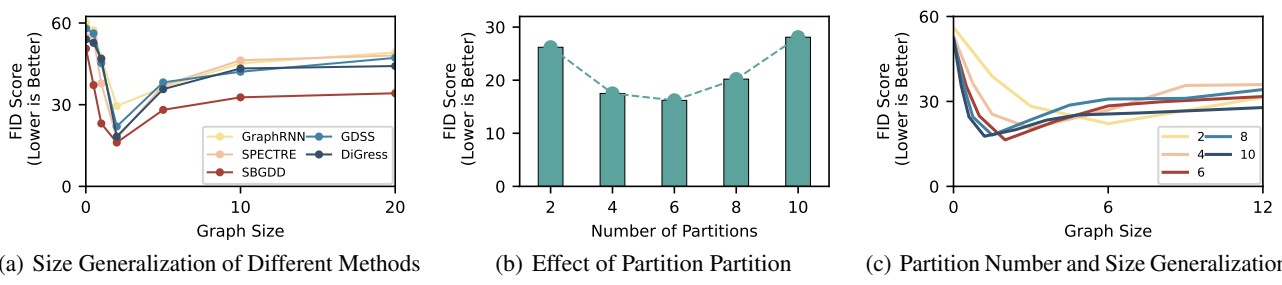

(a) Size Generalization of Different Methods   (b) Effect of Partition Partition   (c) Partition Number and Size Generalization

*Figure 4.* Experiments on Size Generalization and Partition Number. Figure 4(a) illustrates the size generalization performance of different methods. Figure 4(b) shows the performance of our method as a function of the number of partitions. Figure 4(c) demonstrates how the number of partitions impacts the size generalization of our method.

train and learn on the OGBN-products dataset. This validates the effectiveness of our approach in addressing the main motivation for making the graph generation method scalable and efficient for large graphs. In addition, we observe that recent diffusion-based methods such as DiGress consistently outperform other deep-learning based method such as GraphRNN. This is consistent with the existing literature and further indicates our improvement on the diffusion model is important for the graph generation.

**Size Generalization.** Next, we evaluate the size generalization of our method compared to existing baselines. For this, we use the cSBM model and focus on the FID metric to evaluate the distance between the generated distribution and the training distribution. As illustrated in Fig. 3 and 4(a), our method can maintain global structure characteristic and significantly excels in generating graphs with varied sizes. This indicates that our method is able to extrapolate better in size generation with respect to complex/large graph generation, validating our previous theoretical discussion on the benefit of block-representation.

**Partition Number.** Next, we investigate how the partition number, the main hyperparameter of our framework, can affect the learning process. For this, we are interested in two questions: 1) how does the partition number affect the generation quality and 2) how does the partition number affect the size generalization ability of the method? For the first question, we evaluate the FID score (relative to the training dataset) with respect to different partition number. As illustrated in Fig. 4(b), there is an optimal point for the partition number. We hypothesize the reason behind this is that when the partition number is too large (block size being too small), the method would be unable to learn some important global structure. On the other hand, when the partition number is too small (block size being too large), the method would pick up noise from the undesired global structure. Next, we look at how partition number affects the size generalization Based on the previous observation, we would expect that a larger partition number (leading to a smaller block size) would extrapolate better for generating a smaller graph. On the other hand, a lower partition number (leading to a larger block size) would extrapolate better for generating a larger graph. The results in Fig. 4(c) indeed match our expectation.

*Table 2.* Performance Comparison of Graph Generation Methods. The symbol ↓ indicates that a lower value is preferable, while ↑ indicates that a higher value is better. The best and second-best performances are highlighted in dark gray and light gray, respectively. ”N.S.” means the method does not support the dataset,” ”O.O.M.” means ”Out of Memory,” and ”N.A.” means the metric is not available for the dataset. ”M.R.” denotes the memory ratio, with the baseline reference being our method.

| Dataset | Method ↓ | Deg. ↓ | Clus. ↓ | Orbit ↓ | Spec. ↓ | Wavelet ↓ | Avg. ↓ | V.U.N ↑ | FID ↓ | M.R. ↓ |
|---|---|---|---|---|---|---|---|---|---|---|
| Planar Graphs | GraphRNN | 0.097 ± 2.11e-3 | 0.325 ± 1.23e-3 | 0.650 ± 3.25e-3 | 0.933 ± 1.66e-3 | 0.196 ± 1.02e-3 | 0.44 | 53.0 ± 0.795 | 38.0 ± 1.02 | 6.1 |
| | SPECTRE | 0.102 ± 1.07e-3 | 0.252 ± 1.52e-3 | 0.619 ± 2.42e-3 | 0.871 ± 3.42e-3 | 0.176 ± 1.52e-3 | 0.40 | 60.0 ± 0.892 | 33.3 ± 0.842 | 4.2 |
| | EDGE | 0.204 ± 1.12e-3 | 0.522 ± 1.72e-3 | 0.589 ± 2.17e-3 | 0.788 ± 3.21e-3 | 0.154 ± 1.56e-3 | 0.45 | 60.6 ± 1.731 | 46.7 ± 2.768 | 5.3 |
| | EDP-GNN | 0.058 ± 9.01e-4 | 0.648 ± 8.76e-4 | 0.639 ± 1.01e-3 | 0.886 ± 2.83e-3 | 0.170 ± 2.01e-3 | 0.48 | 73.1 ± 0.923 | 36.8 ± 1.391 | 5.5 |
| | GDSS | 0.085 ± 1.43e-3 | 0.275 ± 7.71e-4 | 0.421 ± 1.72e-3 | 0.773 ± 1.87e-3 | 0.098 ± 1.89e-3 | 0.33 | 87.5 ± 2.127 | 33.3 ± 1.721 | 4.8 |
| | DiGress | 0.076 ± 7.58e-4 | 0.260 ± 6.21e-4 | 0.451 ± 5.90e-4 | 0.775 ± 3.32e-3 | 0.094 ± 1.07e-3 | 0.33 | 86.2 ± 3.981 | 29.0 ± 3.011 | 5.7 |
| | SBGD | 0.081 ± 6.21e-4 | 0.216 ± 5.81e-4 | 0.446 ± 1.89e-3 | 0.765 ± 4.21e-3 | 0.093 ± 7.34e-4 | 0.32 | 89.5 ± 1.642 | 25.0 ± 2.857 | 1.0 |
| Contextual Stochastic Block Model | GraphRNN | 0.105 ± 1.78e-3 | 0.388 ± 3.11e-3 | 0.538 ± 1.53e-3 | 0.758 ± 2.46e-3 | 0.210 ± 1.82e-3 | 0.40 | 67.7 ± 2.86 | 29.7 ± 1.28 | 6.0 |
| | SPECTRE | 0.087 ± 5.67e-4 | 0.355 ± 2.43e-3 | 0.499 ± 2.31e-3 | 0.741 ± 2.75e-3 | 0.174 ± 1.42e-3 | 0.37 | 70.3 ± 3.72 | 17.6 ± 0.92 | 4.0 |
| | EDGE | N.S. | N.S. | N.S. | N.S. | N.S. | N.S. | N.S. | N.S. | N.S. |
| | EDP-GNN | N.S. | N.S. | N.S. | N.S. | N.S. | N.S. | N.S. | N.S. | N.S. |
| | GDSS | 0.082 ± 1.43e-3 | 0.298 ± 8.21e-4 | 0.448 ± 7.21e-4 | 0.538 ± 2.32e-3 | 0.164 ± 9.21e-4 | 0.31 | 66.9 ± 5.12 | 22.1 ± 0.88 | 5.5 |
| | DiGress | 0.083 ± 2.08e-3 | 0.302 ± 7.25e-4 | 0.457 ± 6.39e-4 | 0.610 ± 1.59e-3 | 0.165 ± 8.63e-4 | 0.32 | 83.1 ± 1.74 | 18.8 ± 0.53 | 5.3 |
| | SBGD | 0.078 ± 1.00e-3 | 0.287 ± 5.77e-4 | 0.431 ± 3.97e-4 | 0.537 ± 1.25e-3 | 0.154 ± 6.44e-4 | 0.29 | 85.2 ± 3.21 | 16.3 ± 0.32 | 1.0 |
| QM9 | GraphRNN | N.A. | N.A. | N.A. | N.A. | N.A. | N.A. | 75.6 ± 2.54 | 39.5 ± 3.35 | 6.4 |
| | SPECTRE | N.A. | N.A. | N.A. | N.A. | N.A. | N.A. | 68.6 ± 1.88 | 35.5 ± 2.22 | 4.5 |
| | EDGE | N.S. | N.S. | N.S. | N.S. | N.S. | N.S. | N.S. | N.S. | N.S. |
| | EDP-GNN | N.S. | N.S. | N.S. | N.S. | N.S. | N.S. | N.S. | N.S. | N.S. |
| | GDSS | N.A. | N.A. | N.A. | N.A. | N.A. | N.A. | 88.6 ± 2.62 | 21.6 ± 1.56 | 5.7 |
| | DiGress | N.A. | N.A. | N.A. | N.A. | N.A. | N.A. | 92.8 ± 1.42 | 19.7 ± 1.25 | 5.5 |
| | SBGD | N.A. | N.A. | N.A. | N.A. | N.A. | N.A. | 91.2 ± 1.08 | 20.0 ± 1.02 | 1.0 |
| OGBN-Arxiv | GraphRNN | 0.077 ± 1.68e-3 | 0.176 ± 2.31e-3 | 0.382 ± 2.00e-3 | 0.953 ± 1.64e-3 | 0.185 ± 3.18e-3 | 0.35 | N.A. | 33.9 ± 2.13 | 6.2 |
| | SPECTRE | 0.095 ± 2.31e-3 | 0.177 ± 2.71e-3 | 0.371 ± 2.64e-3 | 0.822 ± 1.27e-3 | 0.175 ± 1.94e-3 | 0.33 | N.A. | 25.0 ± 1.54 | 4.3 |
| | EDGE | N.S. | N.S. | N.S. | N.S. | N.S. | N.S. | N.S. | N.S. | N.S. |
| | EDP-GNN | N.S. | N.S. | N.S. | N.S. | N.S. | N.S. | N.S. | N.S. | N.S. |
| | GDSS | 0.066 ± 1.92e-3 | 0.165 ± 1.62e-3 | 0.376 ± 2.27e-3 | 0.789 ± 1.99e-3 | 0.172 ± 1.23e-3 | 0.31 | N.A. | 21.6 ± 1.56 | 5.6 |
| | DiGress | 0.060 ± 1.72e-3 | 0.156 ± 1.22e-3 | 0.285 ± 1.97e-3 | 0.761 ± 2.82e-3 | 0.171 ± 1.68e-3 | 0.28 | N.A. | 33.7 ± 1.21 | 5.3 |
| | SBGD | 0.062 ± 2.01e-3 | 0.148 ± 1.84e-3 | 0.271 ± 1.06e-3 | 0.739 ± 3.48e-3 | 0.173 ± 1.74e-3 | 0.27 | N.A. | 21.4 ± 1.23 | 1.0 |
| OGBN-products | GraphRNN | OOM | OOM | OOM | OOM | OOM | OOM | OOM | OOM | OOM |
| | SPECTRE | OOM | OOM | OOM | OOM | OOM | OOM | OOM | OOM | OOM |
| | EDGE | N.S. | N.S. | N.S. | N.S. | N.S. | N.S. | N.S. | N.S. | N.S. |
| | EDP-GNN | N.S. | N.S. | N.S. | N.S. | N.S. | N.S. | N.S. | N.S. | N.S. |
| | GDSS | OOM | OOM | OOM | OOM | OOM | OOM | OOM | OOM | OOM |
| | DiGress | OOM | OOM | OOM | OOM | OOM | OOM | OOM | OOM | OOM |
| | SBGD | 0.092 ± 8.38e-4 | 0.141 ± 1.56e-3 | 0.467 ± 1.32e-3 | 0.921 ± 2.73e-3 | 0.231 ± 1.80e-3 | 0.37 | N.A. | 18.7 ± 2.78e-3 | 1 |

## 5. Related Works

**Graph Generation.** Graph generation research has a rich history, underscored by the importance of the problem, and is characterized by the use of various random graph models. Notable among these are the Erdos-Renyi (ER) model (Erdős et al., 1960), Barabasi-Albert model (Barabási & Albert, 1999), and the Stochastic-Block Model (SBM) (Holland et al., 1983) , each contributing uniquely to our understanding of graph structures. The SBM, in particular, is pivotal because it builds on the observation that real-life networks often consist of more fundamental block or community structures. This is evident in the way real-life networks exhibit block structures (Su & Marbach, 2022; 2023; Abbe, 2018), where vertices within a block show similar behaviours and have dense interconnections, while inter-block connections remain sparse. Many graph algorithms have been developed based on this understanding, leveraging the inherent block structure in networks. In our current work, we connect this traditional knowledge in graph analysis with the recent advancements in graph diffusion generative models. By applying the block structure prior, we aim to enhance the graph diffusion model's performance, particularly in scaling up to handle larger graphs.

**Graph Diffusion Generative Model.** Inspired by the success of DDM in other domains, there is increasing attention on extending DDM into the graph domains. (Niu et al., 2020) is one of the pioneer works in graph generation that extends the diffusion model in the graph domain. (Niu et al., 2020) shows that a diffusion-based model combined with a graph-neural-like backbone is permutation invariant and capable of capturing the complex distribution of the graph structure of real-life graphs. (Jo et al., 2022) further extend the procedure to incorporate graph structure. In order to speed up the sampling efficiency, (Chen et al., 2023) propose to use Bernoulli distribution as the diffusion process to model the generation and deletion of edges and to use an empty graph as the convergent point. However, their method suffers from two limitations: 1) their methods only model the generation of graph structure and are inapplicable to graphs with features (i.e., inapplicable to most of the interesting applications); and 2) they use a single point in space (empty graph) as the convergent distribution, which seriously limits the expressive power of the generation process (Xu et al., 2022). (Vignac et al., 2022) aims to tackle the discrete nature of graph structure data and extend the discrete diffusion (Hoogeboom et al., 2021; Austin et al., 2021) to the graph structure data with categorical features. Their method also requires accommodating the complete graph in training or sampling, but they are still suffering from the memory explosion problem.

Recent work also explores multi-modal distributions under-

lying graph structures to enhance generation accuracy (Jo et al., 2023). Moreover, an alternative line of research approaches graph generation as an autoregressive editing process on graphs (Zhao et al., 2024), fundamentally distinct from the diffusion framework. An intriguing future research direction would be to investigate potential synergies between these autoregressive and diffusion-based generative paradigms.

# 6. Concluding Discussion

## 6.1. Conclusion

We present the SBGD model, which leverages a block graph representation to address key challenges faced by existing GDGMs. By utilizing a block-based structure, SBGD not only reduces memory complexity, improving scalability, but also enhances size generalization. Through extensive empirical evaluations, we show that SBGD significantly improves memory efficiency and scalability, while demonstrating superior generalization across graphs of varying sizes. This makes SBGD a more flexible and efficient solution to the graph generation problem.

## 6.2. Future Work

Our experiments reveal a trade-off when increasing the number of partitions in the block representation. We hypothesize that the key factor controlling this trade-off is the alignment between the resulting block size, the inherent granularity of the graph structure, and the granularity required for the downstream task. Further investigation into this issue presents an exciting direction for future research. Specifically, developing metrics to quantitatively capture this phenomenon could provide valuable insights, allowing us to refine the block representation for improved performance across a wide range of graph generation tasks.

# Impact Statement

This paper presents work whose goal is to advance the scalability of score-based graph generation method. There are many potential societal consequences of our work, none which we feel must be specifically highlighted here

# Acknowledgement

We would like to thank the anonymous reviewers and area chairs for their helpful comments. This work was supported in part by grants from the National Key Research and Development Program of China (Grant No. 2023YFC3707905), and the Natural Science Foundation of China (No. 42302326).

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

# A. Background

**Denoising Diffusion Models (DDM).** DDM models (Song & Ermon, 2019; Song et al., 2020b; Song & Ermon, 2020; Sohl-Dickstein et al., 2015; Song et al., 2020a) belong to the large family of latent variable models which use a latent space of the same dimension as the data. It models the data generation process as a transition process in the latent space and primarily comprises two main components: a noise model (forward process) and a denoising neural network (backward process). The noise model $q$ gradually corrupts a data point $x$ to form a sequence of increasingly noisy data points $(x_1, \ldots, x_T)$. It adheres to a Markovian structure, formulated as:

$$q(x_1, \ldots, x_T | x) = q(x_1 | x) \prod_{t=2}^{T} q(x_t | x_{t-1}) \tag{A.1}$$

Then, we want to learn a denoising network $\phi_\theta$ that aims to reverse this process by predicting $x_{t-1}$ from $x_t$. To synthesize new samples, a noisy point is sampled from the prior distribution $p(x_T)$ and then inverted through the iterative application of the denoising network. For a diffusion model to exhibit efficiency, it should satisfy three properties:

1. The distribution $q(x_t | x)$ should possess a closed-form equation, facilitating parallel training across varying time steps.

2. The posterior $p_\theta(x_{t-1} | x_t) = \int q(x_{t-1} | x_t, x) d\phi_\theta(x)$ must also be expressed in a closed-form, enabling the utilization of $x$ as the neural network's target.

3. The limit distribution $q_\infty = \lim_{T \to \infty} q(x_T | x)$ must remain independent of $x$, making it viable as a prior distribution for inference.

A scheme that satisfies all these properties is to use Gaussian noise (Ho et al., 2020) as the noise model and it has become the de-facto option for the forward process. This amounts to a Markov chain that gradually adds Gaussian noise to the data according to a variance schedule $\beta_1, \ldots, \beta_T$:

$$q(x_t | x_{t-1}) := \mathcal{N}(x_t; \sqrt{1 - \beta_t} x_{t-1}, \beta_t I), \quad q(x_t | x_0) = \mathcal{N}(x_t; \bar{\alpha}_t x_0, (1 - \bar{\alpha}_t) I), \tag{A.2}$$

where $\alpha_t := 1 - \beta_t$ and $\bar{\alpha}_t := \prod_{s=1}^{t} \alpha_s$. Instead of directly learning a neural net to model the transition from $x_t$ to $x_{t-1}$, one can learn a neural net $f(x_t, t)$ to predict $x_0$ (or $\epsilon$) from $x_t$, and estimate $x_{t-1}$ from $x_t$ and estimated $\hat{x}_0$ (or $\hat{\epsilon}$).

**Graph Diffusion Generative Model(GDGM).** Building upon the empirical success of DDM in other domains, there is increasing attention on extending DDM to tackle the graph generation problem (refer to the related work section for a more detailed discussion). Two notable extensions by (Niu et al., 2020) and (Jo et al., 2022) involve the utilization of matrix representations for graphs, which allows them to treat the graphs in a similar manner as images. However, these approaches exhibit two key limitations: 1) they overlook the discrete nature of graph data, and 2) the dimensionality of the representation space scales as $\mathcal{O}(N^2)$, where $N$ denotes the number of vertices.

To tackle the discrete nature of graph structure data, recent work (Vignac et al., 2022) extends the discrete diffusion model from other domains (Hoogeboom et al., 2021; Austin et al., 2021) and models the graph representation using a state space model, treating the noising process as a Markov transition on the state transitions. However, this approach still suffers from the issue of memory explosion, as each state in the state space model continues to represent the complete graph.

**Block Structure in Graphs.** In real-life graphs, a prominent feature is their block (community structure) (Deshpande et al., 2018; Holland et al., 1983), where vertices within the same block display dense connections and exhibit similar behavioural traits (i.e., similar feature distributions). This contrasts sharply with vertices from different blocks, which are sparsely connected, highlighting the distinct boundaries between these units. The concept of block structure is a cornerstone in graph analysis (Newman et al., 2002; Newman, 2006) and plays a crucial role in many graph learning algorithms such as DeepWalk (Perozzi et al., 2014) and ClusterGCN (Chiang et al., 2019). In this paper, we integrate the traditional principles of graph analysis with the graph diffusion generation model. By leveraging the block structure of the graph as the diffusion space, we aim to significantly reduce memory complexity in our approach.

## A.1. Training Objective Derivation

In this appendix, we present a derivation for the Equations used in the problem formulation for completeness. A similar derivation can be found in (Jo et al., 2022).

The partial score functions can be estimated by training the time-dependent score-based models $s_\theta(.)$ and $s_\phi(.)$, so that

$$s_\theta(\mathcal{G}_t, t) \approx \nabla_{\mathbf{X}} \log \mathbb{P}(\mathcal{G}_t), s_\phi(\mathcal{G}_t, t) \approx \nabla_{\mathbf{A}} \log \mathbb{P}(\mathcal{G}_t).$$

However, the objectives introduced in SGM for estimating the score function are not directly applicable here, since the partial score functions are defined as the gradient of each component, rather than the gradient of the data as in the conventional score function. This interdependence between the two diffusion processes tied by the partial scores adds another layer of difficulty.

To address this issue, an new objective for estimating the partial scores is needed. Intuitively, the score-based models should be trained to minimize the distance to the corresponding ground-truth partial scores. The following new objectives generalize score matching (Song et al., 2020b) to the estimation of partial scores for the given graph dataset, as follows:

$$\min_\theta \mathbb{E}_t \left[ \mathbb{E}_{\mathcal{G}_0} \mathbb{E}_{\mathcal{G}_t|\mathcal{G}_0} \left\| s_{\theta,t}(\mathcal{G}_t) - \nabla_{\mathbf{X}} \log \mathbb{P}(\mathcal{G}_t) \right\|_2^2 \right], \tag{A.3}$$

$$\min_\phi \mathbb{E}_t \left[ \mathbb{E}_{\mathcal{G}_0} \mathbb{E}_{\mathcal{G}_t|\mathcal{G}_0} \left\| s_{\phi,t}(\mathcal{G}_t) - \nabla_{\mathbf{A}} \log \mathbb{P}(\mathcal{G}_t) \right\|_2^2 \right], \tag{A.4}$$

where $t$ is uniformly sampled from $[0, T]$. The expectations are taken over samples $\mathcal{G}_0 \sim p_{\text{data}}$ and $\mathcal{G}_t \sim \mathbb{P}(\mathcal{G}_t|\mathcal{G}_0)$, where $\mathbb{P}(\mathcal{G}_t|\mathcal{G}_0)$ denotes the transition distribution from 0 to $t$ induced by the forward diffusion process.

Unfortunately, the equations above are still not directly trainable since the ground-truth partial scores are not analytically accessible in general. This is why we need to underlying process to be an OU process, as we can leverage the known conditional density of OU process for training.

$$\min_\theta \mathbb{E}_t \left[ \mathbb{E}_{\mathcal{G}_0} \mathbb{E}_{\mathcal{G}_t|\mathcal{G}_0} \left\| s_{\theta,t}(\mathcal{G}_t, t) - \nabla_{\mathbf{X}} \log \mathbb{P}(\mathcal{G}_t|\mathcal{G}_0) \right\|_2^2 \right], \tag{A.5}$$

$$\min_\phi \mathbb{E}_t \left[ \mathbb{E}_{\mathcal{G}_0} \mathbb{E}_{\mathcal{G}_t|\mathcal{G}_0} \left\| s_\phi(\mathcal{G}_t, t) - \nabla_{\mathbf{A}} \log \mathbb{P}(\mathcal{G}_t|\mathcal{G}_0) \right\|_2^2 \right]. \tag{A.6}$$

Since the drift coefficient of the forward diffusion process is linear, the transition distribution $\mathbb{P}(\mathcal{G}_t|\mathcal{G}_0)$ can be separated in terms of $\mathbf{X}_t$ and $\mathbf{A}_t$ as follows:

$$\mathbb{P}(\mathcal{G}_t|\mathcal{G}_0) = \mathbb{P}(\mathbf{X}_t|\mathbf{X}_0) \mathbb{P}(\mathbf{A}_t|\mathbf{A}_0). \tag{A.7}$$

Notably, we can easily sample from the transition distributions of each component, $\mathbb{P}(\mathbf{X}_t|\mathbf{X}_0)$ and $\mathbb{P}(\mathbf{A}_t|\mathbf{A}_0)$, as they are Gaussian distributions with mean and variance determined by the coefficients of the forward diffusion process. This leads to the following training objective:

$$\min_\theta \mathbb{E}_t \left[ \mathbb{E}_{\mathcal{G}_0} \mathbb{E}_{\mathcal{G}_t|\mathcal{G}_0} \left\| s_{\boldsymbol{\theta}}(\mathcal{G}_t, t) - \nabla_{\mathbf{X}} \log \mathbb{P}(\mathbf{X}_t|\mathbf{X}_0) \right\|_2^2 \right], \tag{A.8}$$

$$\min_\phi \mathbb{E}_t \left[ \mathbb{E}_{\mathcal{G}_0} \mathbb{E}_{\mathcal{G}_t|\mathcal{G}_0} \left\| s_{\boldsymbol{\phi}}(\mathcal{G}_t, t) - \nabla_{\mathbf{A}} \log \mathbb{P}(\mathbf{A}_t|\mathbf{A}_0) \right\|_2^2 \right]. \tag{A.9}$$

The expectations in the equation above can be efficiently computed using the Monte Carlo estimate with the samples $(t, \mathcal{G}_0, \mathcal{G}_t)$. Note that estimating the partial scores is not equivalent to estimating $\nabla_{\mathbf{X}} \log \mathbb{P}(\mathbf{X}_t)$ or $\nabla_{\mathbf{A}} \log \mathbb{P}(\mathbf{A}_t)$, the main objective of previous score-based generative models, since estimating the partial scores requires capturing the dependency between $\mathbf{X}_t$ and $\mathbf{A}_t$ determined by the joint probability through time.

### A.1.1. DERIVATION OF TRAINING OBJECTIVE A.3

The original score matching objective can be written as follows:

$$\mathbb{E}_{\mathcal{G}_t} \left[ \| s_{\boldsymbol{\theta}}(\mathcal{G}_t, t) - \nabla_{\mathbf{X}} \log \mathbb{P}(\mathcal{G}_t) \|_2^2 \right] = \mathbb{E}_{\mathcal{G}_t} \left[ \| s_{\boldsymbol{\theta}}(\mathcal{G}_t, t) \|_2^2 \right] - 2\mathbb{E}_{\mathcal{G}_t} \left[ \langle s_{\boldsymbol{\theta}}(\mathcal{G}_t, t), \nabla_{\mathbf{X}} \log \mathbb{P}(\mathcal{G}_t) \rangle \right] + C_1,$$

where $C_1$ is a constant that does not depend on $\mathbf{W}$. On the other hand, we have

$$\mathbb{E}_{\mathcal{G}_t}\mathbb{E}_{\mathcal{G}_t|\mathcal{G}_0}\left[\|s_{\boldsymbol{\theta}}(\mathcal{G}_t,t) - \nabla_{\mathbf{X}}\log\mathbb{P}(\mathcal{G}_t|\mathcal{G}_0)\|_2^2\right] = \mathbb{E}_{\mathcal{G}_t}\mathbb{E}_{\mathcal{G}_t|\mathcal{G}_0}\left[\|s_{\boldsymbol{\theta}}(\mathcal{G}_t,t)\|_2^2\right] - 2\mathbb{E}_{\mathcal{G}_t}\mathbb{E}_{\mathcal{G}_t|\mathcal{G}_0}\left[\langle s_{\boldsymbol{\theta}}(\mathcal{G}_t,t), \nabla_{\mathbf{X}}\log\mathbb{P}(\mathcal{G}_t|\mathcal{G}_0)\rangle\right] + C_2,$$

For the second term, from the derivation (Appendix A.1 from (Jo et al., 2022)), we know that it has the following equivalency:

$$\mathbb{E}_{\mathcal{G}_t}\left[\langle s_{\boldsymbol{\theta}}(\mathcal{G}_t,t), \nabla_{\mathbf{X}}\log\mathbb{P}(\mathcal{G}_t)\rangle\right] = \mathbb{E}_{\mathcal{G}_t}\mathbb{E}_{\mathcal{G}_t|\mathcal{G}_0}\left[\langle s_{\boldsymbol{\theta}}(\mathcal{G}_t,t), \nabla_{\mathbf{X}}\log\mathbb{P}(\mathcal{G}_t|\mathcal{G}_0)\rangle\right]$$

Since the constant $C_1$ and $C_2$ does not affect the optimization results, we can conclude that the following two objectives are equivalent with respect to $\boldsymbol{\theta}$

$$\mathbb{E}_{\mathcal{G}_t}\mathbb{E}_{\mathcal{G}_t|\mathcal{G}_0}\left[\|s_{\boldsymbol{\theta}}(\mathcal{G}_t,t) - \nabla_{\mathbf{X}}\log\mathbb{P}(\mathcal{G}_t|\mathcal{G}_0)\|_2^2\right]$$
$$\mathbb{E}_{\mathcal{G}_t}\left[\|s_{\boldsymbol{\theta}}(\mathcal{G}_t,t) - \nabla_{\mathbf{X}}\log\mathbb{P}(\mathcal{G}_t)\|_2^2\right]$$

Similarly, computing the gradient with respect to $\mathbf{A}$, we can show that the following two objectives are also equivalent with respect to $\boldsymbol{\phi}$:

$$\mathbb{E}_{\mathcal{G}_t}\mathbb{E}_{\mathcal{G}_t|\mathcal{G}_0}\left[\|s_{\boldsymbol{\phi}}(\mathcal{G}_t,t) - \nabla_{\mathbf{A}}\log\mathbb{P}(\mathcal{G}_t|\mathcal{G}_0)\|_2^2\right]$$
$$\mathbb{E}_{\mathcal{G}_t}\left[\|s_{\boldsymbol{\phi}}(\mathcal{G}_t,t) - \nabla_{\mathbf{A}}\log\mathbb{P}(\mathcal{G}_t)\|_2^2\right]$$

Now, it remains to show that $\nabla_{\mathbf{X}}\log\mathbb{P}(\mathcal{G}_t|\mathcal{G}_0)$ is equivalent to $\nabla_{\mathbf{X}}\log\mathbb{P}(\mathbf{X}_t|\mathbf{X}_0)$. Using the chain rule, we get that

$$\frac{\partial\log\mathbb{P}(\mathbf{A}_t|\mathbf{A}_0)}{\partial(\mathbf{X}_t)_{ij}} = \mathrm{Tr}\left[\nabla_{\mathbf{A}}\log\mathbb{P}(\mathbf{A}_t|\mathbf{A}_0)\frac{\partial\mathbf{A}_t}{\partial(\mathbf{X}_t)_{ij}}\right] = 0.$$

With this result, we have that,

$$\nabla_{\mathbf{X}}\log\mathbb{P}(\mathcal{G}_t|\mathcal{G}_0) = \nabla_{\mathbf{X}}\log\mathbb{P}(\mathbf{X}_t|\mathbf{X}_0) + \nabla_{\mathbf{X}}\log\mathbb{P}(\mathbf{A}_t|\mathbf{A}_0) = \nabla_{\mathbf{X}}\log\mathbb{P}(\mathbf{X}_t|\mathbf{X}_0).$$

Therefore, we can conclude that

$$\nabla_{\mathbf{X}}\log\mathbb{P}(\mathcal{G}_t|\mathcal{G}_0) = \nabla_{\mathbf{X}}\log\mathbb{P}(\mathbf{X}_t|\mathbf{X}_0)$$

With a similar computation for $\mathbf{A}_t$, we can also show that $\nabla_{\mathbf{A}}\log\mathbb{P}(\mathcal{G}_t|\mathcal{G}_0)$ is equal to $\nabla_{\mathbf{A}}\log\mathbb{P}(\mathbf{A}_t|\mathbf{A}_0)$.

## B. Algorithm Summarization

SBGD decompose the graphs into finer structural structures (blocks) and operates on a latent graph space consisting of the building block of the graph structure, allowing for the computation of various graph descriptors at each diffusion step. The procedure of training and sampling from SBGD are summarized in Algorithm 1 and Algorithm 2 respectively.

**Implementation.** SBGD decompose the graphs into finer structural structures (blocks) and operates on a latent graph space consisting of the building block of the graph structure, allowing for the computation of various graph descriptors at each diffusion step. The detailed and complete procedure of training and sampling of SBGD are provided in the supplementary material

## C. Graph Partition Problem

The *graph partition problem* is a fundamental problem in graph theory and computer science. It involves dividing a graph into multiple smaller subgraphs, or "partitions," that satisfy certain properties. This problem has wide-ranging applications, including parallel computing, clustering, network analysis, and community detection. The goal is to achieve a division of the graph that satisfies specific constraints while optimizing one or more criteria, such as minimizing the number of edges cut between partitions or ensuring that each partition is balanced in size.

---

**Algorithm 1** SBGD Training Algorithm

---

**Require:** A set of block graphs and their interaction $\mathcal{C} = \{\mathcal{C}_i\}$, $\mathbf{\Delta} = \{\mathbf{A}_{ij}\}$

1: Sample two block graphs $\mathcal{C}_i, \mathcal{C}_j$ from $\mathcal{C}$
2: Extract the corresponding interactions $\mathbf{A}_{ij}$
3: Sample $t \sim \text{Uniform}(1, \ldots, T)$
4: Sample a noise $\epsilon \sim \mathbb{N}(0, \mathbf{I})$
5: Corrupt data:
6: $\quad \mathbf{A}_i^{(t)} = \sqrt{\gamma(t)} \cdot \mathbf{A}_i^{(0)} + \sqrt{1 - \gamma(t)} \cdot \varepsilon$
7: $\quad \mathbf{A}_j^{(t)} = \sqrt{\gamma(t)} \cdot \mathbf{A}_j^{(0)} + \sqrt{1 - \gamma(t)} \cdot \varepsilon$
8: $\quad \mathbf{X}_i^{(t)} = \sqrt{\gamma(t)} \cdot \mathbf{X}_i^{(0)} + \sqrt{1 - \gamma(t)} \cdot \varepsilon$
9: $\quad \mathbf{X}_j^{(t)} = \sqrt{\gamma(t)} \cdot \mathbf{X}_j^{(0)} + \sqrt{1 - \gamma(t)} \cdot \varepsilon$
10: Extract structural and spectral features from the block graph:
$\quad \mathbf{z}_i^{(t)}, \mathbf{z}_j^{(t)} = f(\mathbf{A}_i^{(t)}, \mathbf{X}_i^{(t)}, t), f(\mathbf{A}_j^{(t)}, \mathbf{X}_j^{(t)}, t)$
11: Predict and compute the weighted reconstruction loss:
$\quad \widehat{\mathbf{A}}_i^{(0)} = s_{\boldsymbol{\theta}}(\mathbf{A}_i^{(t)}, \mathbf{X}_i^{(t)}, \mathbf{z}_i^{(t)}, t)$
$\quad \widehat{\mathbf{A}}_j^{(0)} = s_{\boldsymbol{\theta}}(\mathbf{A}_j^{(t)}, \mathbf{X}_j^{(t)}, \mathbf{z}_j^{(t)}, t)$
$\quad \widehat{\mathbf{X}}_i^{(0)} = s_{\boldsymbol{\psi}}(\mathbf{A}_i^{(t)}, \mathbf{X}_i^{(t)}, \mathbf{z}_i^{(t)}, t)$
$\quad \widehat{\mathbf{X}}_j^{(0)} = s_{\boldsymbol{\psi}}(\mathbf{A}_j^{(t)}, \mathbf{X}_j^{(t)}, \mathbf{z}_j^{(t)}, t)$
$\quad \widehat{\mathbf{A}}_{ij} = s_{\boldsymbol{\phi}}(\widehat{\mathbf{A}}_i, \widehat{\mathbf{A}}_j, \widehat{\mathbf{X}}_i, \widehat{\mathbf{X}}_j)$
$\quad \mathcal{L} = \sum_A \mathcal{L}_{\mathbf{A}}(\widehat{\mathbf{A}}^{(0)}, \mathbf{A}^{(0)}) + \sum_X \mathcal{L}_{\mathbf{X}}(\widehat{\mathbf{X}}^{(0)}, \mathbf{X}^{(0)}) + \mathcal{L}(\widehat{\mathbf{A}}_{ij}, \mathbf{A}_{ij})$

---

**Algorithm 2** SBGD sampling algorithm.

---

1: Sample $n_1, n_2$ from the training data distribution
2: Sample $\mathbf{A}_i^{(T)}, \mathbf{A}_j^{(T)}$ from $\mathbb{P}_{\mathbf{A}_{\text{intra}}}(n_1), \mathbb{P}_{\mathbf{A}_{\text{intra}}}(n_2)$
3: Sample $\mathbf{X}_i^{(T)}, \mathbf{X}_j^{(T)}$ from $\mathbb{P}_{\mathbf{X}}(n_1), \mathbb{P}_{\mathbf{X}}(n_2)$
4: **for** $t = T$ to $1$ **do**
5: $\quad \mathbf{z}_i^{(t)}, \mathbf{z}_j^{(t)} = f(\mathbf{A}_i^{(t)}, \mathbf{X}_i^{(t)}, t), f(\mathbf{A}_j^{(t)}, \mathbf{X}_j^{(t)}, t)$
$\quad\quad \widehat{\mathbf{A}}_i = s_{\boldsymbol{\theta}}(\mathbf{A}_i^{(t)}, \mathbf{X}_i^{(t)}, \mathbf{z}_i^{(t)}, t)$
$\quad\quad \widehat{\mathbf{A}}_j = s_{\boldsymbol{\theta}}(\mathbf{A}_j^{(t)}, \mathbf{X}_j^{(t)}, \mathbf{z}_j^{(t)}, t)$
$\quad\quad \widehat{\mathbf{X}}_i = s_{\boldsymbol{\psi}}(\mathbf{A}_i^{(t)}, \mathbf{X}_i^{(t)}, \mathbf{z}_i^{(t)}, t)$
$\quad\quad \widehat{\mathbf{X}}_j = s_{\boldsymbol{\psi}}(\mathbf{A}_j^{(t)}, \mathbf{X}_j^{(t)}, \mathbf{z}_j^{(t)}, t)$
$\quad\quad \widehat{\mathbf{A}}_i^{(t-1)}, \widehat{\mathbf{A}}_j^{(t-1)}, \widehat{\mathbf{X}}_i^{(t-1)}, \widehat{\mathbf{X}}_j^{(t-1)} = \text{DDIM or DDPM}(\widehat{\mathbf{A}}_i, \widehat{\mathbf{A}}_j, \widehat{\mathbf{X}}_i, \widehat{\mathbf{X}}_j, \mathbf{A}_i^{(t)}, \mathbf{A}_j^{(t)}, \mathbf{X}_i^{(t)}, \mathbf{X}_j^{(t)}, t, t-1)$
6: **end for**
7: $\widehat{\mathbf{A}}_{ij} = s_{\boldsymbol{\phi}}(\widehat{\mathbf{A}}_i, \widehat{\mathbf{A}}_j, \widehat{\mathbf{X}}_i, \widehat{\mathbf{X}}_j)$

---

## C.1. Formal Definition of the Graph Partition Problem

Formally, the graph partition problem can be defined as follows: Given a graph $G = (V, E)$, where $V$ is the set of vertices (nodes) and $E$ is the set of edges (connections between nodes), the objective is to partition $V$ into $k$ disjoint subsets $V_1, V_2, \ldots, V_k$ such that:

$$V = V_1 \cup V_2 \cup \cdots \cup V_k, \quad V_i \cap V_j = \emptyset \quad \text{for all} \quad i \neq j$$

Each partition $V_i$ represents a subgraph of $G$, and the partitions are subject to various optimization criteria. A common objective is to minimize the *cut size*, which is the number of edges connecting vertices in different partitions. The cut size is defined as:

$$\text{Cut}(V_1, V_2, \ldots, V_k) = \sum_{1 \leq i < j \leq k} |E(V_i, V_j)|$$

where $E(V_i, V_j)$ is the set of edges between partitions $V_i$ and $V_j$, and $|E(V_i, V_j)|$ is the number of such edges. This is typically referred to as the *min-cut* problem, where the objective is to minimize the number of edges between partitions while ensuring the constraints on the partitioning of the vertices are satisfied.

Beyond the cut size, additional constraints may be imposed on the partitioning, such as balancing the size of each partition (ensuring that the number of vertices in each partition is roughly equal), or optimizing a more complex objective such as minimizing the conductance or maximizing modularity in network analysis.

## C.2. Approaches to Solving the Graph Partition Problem

The graph partition problem, particularly the min-cut problem, is NP-hard, meaning that no efficient algorithm guarantees an optimal solution in polynomial time for all instances. Consequently, various heuristic and approximation methods are employed to tackle the problem. Key approaches include:

- **Spectral Partitioning:** This approach uses the eigenvalues and eigenvectors of the graph's Laplacian matrix to determine the best way to partition the graph. By computing the Fiedler vector (the eigenvector corresponding to the second smallest eigenvalue of the Laplacian matrix), the graph can be split into two parts such that the cut size is minimized. This approach is widely used in clustering and graph partitioning, particularly for large graphs (Chung, 1997; Fiedler, 1973).

- **Optimization-Based Methods:** These methods formulate the graph partitioning problem as a mathematical optimization problem and solve it using techniques such as linear programming, semidefinite programming, or integer programming. While these methods can provide exact solutions, they often suffer from high computational complexity, especially for large graphs (Korte et al., 2011).

- **Community Detection Algorithms:** In the context of networks and social graphs, community detection algorithms aim to partition the graph in such a way that the number of edges within each partition is maximized while the number of edges between partitions is minimized. Methods like the *Louvain algorithm* and *Infomap* have become popular for detecting communities in large-scale networks (Aref & Mostajabdaveh, 2024).

In this paper, we focus on incorporating structural priors into the score-based graph generative model. For the partitioning algorithm, we adopt the commonly used METIS algorithm (Karypis & Kumar, 1998) from the DGL library (Wang et al., 2019). Further exploration of partition algorithm selection and its impact on the performance of SGBD presents an exciting direction for future research.

We select the METIS partitioning algorithm primarily due to our core modeling assumption, which posits dense and homogeneous intra-block (diagonal) structures coupled with sparse inter-block (off-diagonal) connections. METIS excels in partitioning graphs into balanced, cohesive clusters while simultaneously minimizing edge cuts. This characteristic aligns closely with our scenario, making METIS a particularly suitable choice for capturing and leveraging the inherent block-wise structure of the graphs under investigation.

# D. Experiment Details

## D.1. General Setup

**Baselines.** In our experiments, we compare the performance of SBGD against several state-of-the-art denoising-diffusion-based graph generation methods including DiGress (Vignac et al., 2022), GDSS (Jo et al., 2022), and EDP-GNN (Niu et al., 2020). In addition, we also consider several representative deep graph generation such as GraphRNN (You et al., 2018), SPECTRE (Martinkus et al., 2022), and EDGE (Chen et al., 2023). The summarized description of each baseline are as follows.

- **GraphRNN**: A pioneering autoregressive model for graph generation that sequentially generates graphs by adding nodes and edges in a breadth-first-search order, effectively capturing the structure of small- to medium-sized graphs.

- **SPECTRE**: A graph generation framework leveraging spectral decomposition to generate realistic graph structures, focusing on preserving spectral properties like eigenvalues and eigenvectors for structural fidelity.

- **EDGE**: An efficient graph generation method that employs generate graph edge-by-edge, enabling scalability.

- **EDP-GNN**: A pioneering diffusion-based graph generation model for generating graph structure.

- **DiGress**: A state-of-the-art denoising diffusion-based framework based on Markov state model.

- **GDSS**: A graph generation method utilizing score-based generative modeling with stochastic differential equations (SDEs) to model the gradient field of the graph distribution.

**Datasets.** We consider five real and synthetic datasets with varying sizes and connectivity levels: Planar-graphs, Contextual Stochastic Block Model(cSBM) (Deshpande et al., 2018), Proteins (Dobson & Doig, 2003), QM9 (Wu et al., 2018), OGBN-Arxiv, and OGBN-Products (Hu et al., 2021). Notably, OGBN-Products is a relatively large dataset for graph generation tasks, and our method is the only approach capable of successfully training on it. The summarized descriptions of each datasets are as follow.

- **Planar-graphs**: A synthetic dataset consisting of planar graphs, where nodes and edges are arranged such that they can be embedded in the plane without edge crossings. This dataset is used to evaluate models on generating graphs with specific structural constraints.

- **cSBM**: cSBM is an extension of the classic Stochastic Block Model (SBM), designed to incorporate node features or context into the synthetic.

- **Proteins**: A real-world dataset consisting of protein structures represented as graphs, where nodes correspond to amino acids and edges represent spatial or functional interactions.

- **QM9**: A dataset of small molecules represented as graphs, where nodes represent atoms and edges represent bonds.

- **OGBN-Arxiv**: A large citation network dataset where nodes represent papers and edges denote citation relationships. Node features are derived from paper abstracts.

- **OGBN-Products**: A large-scale product co-purchasing network where nodes represent products and edges signify co-purchase relationships.

## D.2. Model Description and Configuration

For our implementation, we follow the approach outlined in (Vignac et al., 2022) and utilize a Graph Transformer as the score network. Graph Transformers are a class of models that adapt the transformer architecture, originally designed for sequential data, to process graph-structured data. These models are particularly well-suited for tasks such as node classification, link prediction, and graph generation. One of the main advantages of Graph Transformers over traditional Graph Neural Networks (GNNs) is their ability to model long-range dependencies in graph data. By leveraging the attention mechanism, Graph Transformers can capture relationships between distant nodes, without requiring explicit graph convolutions, thus offering a more flexible and scalable approach to graph representation learning.

### D.3. Hyperparameter Tuning and Training

For training our network, we adopt the widely-used Adam optimizer, tuning only the learning rate as the primary hyperparameter. To determine the optimal values for other hyperparameters in our model, we perform a simple grid search over the following ranges:

- **Number of layers:** [2, 4]

- **Hidden dimension:** [8, 16, 32, 64, 128, 256]

- **Learning rate:** [0.1, 0.05, 0.01, 0.005, 0.001]

- **Diffusion Length $T$:** [50,100,200]

- **Sampling Steps:** [100,200,500,1000]

For the variance schedule, we follow the one in (Jo et al., 2022).

### D.4. Evaluation Metric

Our evaluation focuses on two main aspects: (1) the quality of the generated graphs, assessing how well the method captures the underlying graph distribution, and (2) the memory consumption required during graph generation. To assess the quality of the generated graphs, we follow established graph generation studies and adopt both structure-based and neural-based metrics. These metrics allow us to evaluate the generated graphs from different perspectives, ensuring a comprehensive comparison across methods.

#### D.4.1. STRUCTURE-BASED METRICS

Structure-based metrics evaluate how well the generated graphs match the statistical properties of real-world graphs. These metrics are critical for understanding whether the generated graphs preserve important structural characteristics like node connectivity and community structure, which are crucial for tasks like graph analysis and network modeling.

**Maximum Mean Discrepancy (MMD)**: MMD is a kernel-based metric that quantifies the difference between the distribution of graph properties in the generated graph and the real graph. Specifically, MMD measures how well the generated graph approximates the distribution of properties like node degrees, clustering coefficients, and orbit counts. The degree distribution reflects how nodes are connected within the graph, the clustering coefficient measures the tendency of nodes to form local clusters, and orbit counts capture the higher-order relationships between nodes (such as triangles and motifs). By comparing these properties between the generated and real graphs, MMD provides a robust measure of structural fidelity, where a lower MMD indicates better alignment between the graphs (Gretton et al., 2012).

#### D.4.2. NEURAL-BASED METRICS

Neural-based metrics evaluate the quality of generated graphs in the context of learned representations. These metrics are useful for assessing whether the generator has captured not only low-level structural properties but also more abstract features that are harder to quantify with traditional metrics. This allows us to measure the generative model's performance in a more holistic manner.

**Fréchet Inception Distance (FID)**: FID is a widely used metric in generative models, originally proposed for image generation, that compares the distribution of features extracted from a learned embedding space between the real and generated graphs. Specifically, we use FID to assess the alignment between graph distributions in the embedding space produced by a pre-trained neural network. The key advantage of FID is that it compares high-level feature representations, providing insights into whether the generative model produces graphs that are not only structurally similar but also preserve the higher-level semantic relationships. A lower FID score indicates that the generated graphs are more similar to the real graphs in the learned embedding space (Heusel et al., 2017). This is particularly important when we are dealing with complex graph data that may contain non-trivial patterns or higher-order structures that are not easily captured by traditional structural metrics.

### D.4.3. MEMORY CONSUMPTION METRICS

In addition to evaluating the quality of the generated graphs, we also focus on the memory efficiency of the graph generation process. Memory consumption is a crucial factor, especially for large-scale graph generation tasks, as excessive memory usage can hinder the scalability of the model.

**Memory Consumption Ratio**: To ensure a direct comparison of memory efficiency between different methods, we define the memory consumption ratio as the ratio of the baseline model's memory consumption to our model's memory consumption. This metric highlights the relative efficiency of our approach in terms of memory usage during graph generation. A lower ratio indicates that our method requires less memory, making it more scalable and efficient, particularly when generating large graphs. By using our method as a reference point, we provide a clear comparison of how different approaches perform in terms of memory consumption, which is essential for practical applications that require handling large-scale graphs.

