# OpenReview forum: "SBGD: Improving Graph Diffusion Generative Model via Stochastic Block Diffusion"
_ICML.cc/2025/Conference — ICML 2025 poster_

### Official Review · Reviewer_mdPb · 2025-03-07

**Overall Recommendation:** 3

**Summary:**

This paper introduces the SBGD model to address the scalability and size generalization challenges of Graph Diffusion Generative Models (GDGMs). Traditional GDGMs struggle with high memory requirements and poor generalization to graph sizes not seen during training. SBGD mitigates these issues by refining graph representations into a block graph space, leveraging structural priors to reduce memory complexity and enhance generalization. Empirical evaluations demonstrate that SBGD achieves memory savings while maintaining the graph generation performance of state-of-the-art methods.

**Claims And Evidence:**

The proposed SBGD method claims improved computational efficiency and generalizability. While the authors provide complexity comparisons in Table 1 and memory ratios in Table 2, an empirical analysis of training time would strengthen their efficiency claims. Regarding generalizability, the term "graph size" is somewhat ambiguous. In Figure 3, it appears to refer to the density of the edges. Please clarify the definition of "graph size."

**Essential References Not Discussed:**

See Experimental Designs Or Analyses for DruM[1].

[1] Graph Generation with Diffusion Mixture, ICML 2024

**Experimental Designs Or Analyses:**

The experiments are well-designed and effectively support the claims of efficiency and generalizability. One thing to note is that the authors missed one state-of-the-art baseline DruM[1]. Please add that into comparison if possible.

[1] Graph Generation with Diffusion Mixture, ICML 2024

**Methods And Evaluation Criteria:**

The method is straightforward yet appears effective. The decomposition of large graphs into blocks facilitates sparse graph construction. The efficiency and effectiveness of the proposed model are closely tied to how the graph is partitioned. The authors address this in Figure 4 with empirical results. How do the authors determine the number of partitions? Is it treated as a hyperparameter, or is it based on another metric?

In Appendix C2, the authors use METIS partitioning, which results in balanced graph blocks with approximately equal node counts. This might not be the case for real graphs, while we might find unbalanced blocks. Did the authors compare this method with other partitioning techniques?

**Other Comments Or Suggestions:**

See above.

**Other Strengths And Weaknesses:**

See above.

**Questions For Authors:**

The authors mention distributed training for block-wise modeling in their theoretical discussion, which is somewhat unconvincing. Node-wise samplers are well-implemented in both DGL and PYG and support distributed training. Additionally, treating blocks as batches may lead to information loss if edges between blocks are ignored. Can the authors elaborate on this?

**Relation To Broader Scientific Literature:**

Graph generation is a significant issue with applications across various domains. Diffusion-based graph generative models struggle with large graphs, and the proposed SBGD addresses this through block-wise modeling. While this contribution is noteworthy, a more in-depth discussion of its potential positive and negative impacts would be beneficial.

**Theoretical Claims:**

The authors provide a theoretical discussion on model complexity. It would be beneficial to include more details on how these results are derived.

---

> ### Author Rebuttal · Authors · 2025-04-01
>
> Dear reviewer, thank you for the insightful comments and helpful suggestions. These greatly help us improve upon the current paper, and we appreciate the opportunity for addressing your questions and concerns here.
>
> ## Clarification of Graph Size
>
> We intended the term "graph size" to specifically refer to the number of nodes. The confusion may stem from the visualizations in Figure 3, where denser layouts naturally arise as the node count increases (given a fixed edge-to-node ratio), making the graphs appear more connected. We will revise the manuscript to make this definition explicit and improve the accompanying figure captions for clarity.
>
> ## Determination of the Number of Partitions
>
> The number of partitions is treated as a tunable hyperparameter. As shown in Figure 4(b), our empirical analysis suggests that the optimal number of partitions depends on both the dataset and task. Extremely fine or coarse partitions tend to degrade performance due to underfitting or oversmoothing, respectively. We will explicitly state this in the revised manuscript and offer guidance on selecting this hyperparameter based on validation performance.
>
> ## Comparison with Other Partitioning Techniques
>
> Our use of the METIS partitioning algorithm is motivated by the core modeling assumption that intra-block (diagonal) structures are dense and relatively homogeneous, while inter-block (off-diagonal) interactions are sparse. METIS is well-suited for this setting, as it efficiently divides graphs into balanced clusters while minimizing edge cuts.
>
> We did experiment with random partitioning during early development, but observed significantly worse performance—likely due to incoherent block structures and noisier learning dynamics. We will aim to include a more systematic study of this comparison in the revised version.
>
> ## Details on Theoretical Derivation
>
> Thank you for pointing this out. We agree that providing more detailed derivation steps will enhance the clarity and completeness of the theoretical section. We will expand this part in the revision to include step-by-step derivations and clarify underlying assumptions.
>
> ## Comparison with DruM (ICML 2024)
>
> We appreciate the pointer to DruM—an interesting and related work. While both papers share a modular principle, the core assumptions and objectives differ:
>
> - DruM assumes that the graph distribution is multi-modal, and it decomposes the generative process into learning distinct modes to facilitate inference.
> - In contrast, our approach focuses on structural decomposition for memory efficiency, leveraging graph partitioning to operate on block-level representations. Our method does not rely on mixture modeling of distributions and avoids storing the full graph at once.
>
> We will cite DruM in the revised manuscript and include a more thorough discussion in the related work section to highlight the differences and complementarity.
>
> ## Discussion on Broader Impacts
>
> Thank you for noting the omission of an impact statement. We will include one in the revised manuscript. Our method is designed for memory-efficient graph generation, which could lower the barrier for training generative models in resource-constrained environments, such as laboratory settings or edge devices. We do not foresee any immediate or specific negative societal impacts, but will include a balanced impact statement in line with standard publication policies.
>
> ## Clarification on Distributed Training and Information Loss
>
> We appreciate this question. In our training procedure, the fundamental unit is a **block pair**, which consists of two block graphs and their mutual interactions. In addition to training score networks for each block, we train a lightweight interaction network to model the off-diagonal connections.
>
> This design enables distributed training, where block pairs can be allocated across multiple devices without the need for inter-device communication, since each block pair is self-contained. We will clarify this in the revised manuscript to highlight the scalability benefits of our approach.
>
> Once again, we sincerely thank the reviewer for the thoughtful and constructive feedback. We hope our response has satisfactorily addressed your questions and concerns.

---

> > ### Comment · Reviewer_mdPb · 2025-04-04
> >
> > Thank you for the rebuttal. I will keep my score to support this work.

---

> > > ### Author Response · Authors · 2025-04-04
> > >
> > > Thank you very much for your support and for taking the time to consider our rebuttal. We truly appreciate your constructive feedback and encouragement.

---

### Official Review · Reviewer_KH7S · 2025-03-10

**Overall Recommendation:** 4

**Summary:**

The manuscript introduces SBGD, a graph diffusion generative model based on a block representation inductive bias and aiming for lower memory complexity in the large-graph limit. SBGD first partitions the graph's nodes into $k$ non-overlapping groups using some pre-determined non-learned algorithm (here METIS). These groups partition the adjacency matrix into blocks of edges within the same group (diagonal blocks) or between two different groups (off-diagonal blocks). The Markovian forward process is decomposed into three independent "noising" conditional distribution: noising the node features $X_{i}^{(t)}$ group $i$'s nodes, noising the $i$-th diagonal block $A_{i}^{(t)}$ or noising the off-diagonal block $A_{ij}^{(t)}$ between group $i$ and group $j$. The noising and learned network score functions for the features and diagonal blocks are very Digress-like, and something lighter (not 100% clear to me at this time) is done for the off-diagonal blocks. The (sampling) backward process for each group is independent (though node features and diagonal blocks are jointly generated), and off-diagonal blocks are generated in view of the two groups they connect (again, details not 100% clear). Empirical results appear good. The theoretical scaling complexity is improved, and the manuscript's algorithm can run on OGBN-products while other approaches OOM on the same (frugal-ish) hardware.

**Claims And Evidence:**

(Claim numbering is my own.)

### Claim 1

> Empirical results show that SBGD achieves significant memory improvements (up to 6×)

The core idea of breaking down a large graph into block structures to improve memory efficiency makes a lot of sense, and this claim is supported both theoretically (Section 3.3 and Table 1) and experimentally (last column of Table 2).

### Claim 2

> while maintaining comparable or even superior graph generation performance relative to state-of-the-art methods.

The results presented in Table 2 support this claim that the method offers competitive performances despite the memory savings. I have reservations as to how performances on this type of benchmark would translate to concrete use-case scenario, but this is an issue pervading the whole generative graph modeling subfield: I don't think that I can fault the manuscript for this.

### Claim 3

> experiments on size generalization demonstrate that SBGD exhibit better size generation, particularly exceling at generating graphs larger than those seen in the training set

I fail to find the experimental setting details for the results shown in Figures 3 and 4(a). Figure 3 is not convincing to me: the training graph (leftmost) in Figure 3 has block-like structure quite aligned with the manuscript's main new modeling assumption, and if similar graphs are used in Figure 4(a), then the manuscript's claim is greatly weakened, and should minimally be amended to clarify this reliance on very specific graph structures.

### Claim 4

> that smaller block representations initially improve performance, but excessively small blocks can degrade generative quality

The claim makes sense and is supported by Figures 4(b-c), although it is not clear what dataset/protocol was used to produce these figures. The suggestion of "an optimal block size, with granularity depending on the data’s properties" could have (but wasn't) easily probed by plotting Figures 4 (b-c) for different datasets.

### Claim 5

> it exemplifies the principle of modularization in generative modeling, offering a novel way to explore generative models by decomposing complex tasks into more manageable components

It does. Writing this review spurred many ideas as to how to "fix" this work's weakest points, ideas made "obvious" now that this corner of the design space has been made visible to me. This manuscript has faults, but its core idea is both valid and important.

**Essential References Not Discussed:**

See above.

**Experimental Designs Or Analyses:**

I already mentioned issues surrounding Figures 3 and 4.

I am not anymore following this subfield in enough details to properly assess the details surrounding Table 2.

**Methods And Evaluation Criteria:**

As mentioned above, there appear to be no details (neither in the main paper nor appendices) as to the experimental setting surrounding Figures 3 and 4.

Also as mentioned above, to the best of my understanding, the evaluation criteria are limited but standard for the subfield.

I believe that the presentation of SBGD lacks important details necessary for reproducibility, notably on the front of the off diagonal blocks. In particular, lines 8 and 10 of Algorithm 2 (Appendix B) are incomplete (and/or incompatible). Is $\widehat{A}_{12}$ obtained in one shot, or by DDIM/DDPM? This needs to be clear!

Moreover, my understanding is that lines 6 and 7 of the same Algorithm 2 should both involve two separate calls to the score functions (like is done for the features on line 5), and that what is currently displayed on those lines implies a coupling between the diagonal blocks that is not mentioned elsewhere in the manuscript. Also, compared to Algorithm 1, the explicit dependencies in features are gone.

This appendix appears to aim for conciseness, at the cost of clarity/correctness, but there is no space limit for appendices. Please be clear!

**Other Comments Or Suggestions:**

The manuscript favours an "assortative" view, where the off-diagonal blocks are sparser than the diagonal ones. But disassortative networks exist, and some generalization of this model could address them.

More generally, I suggest that the authors ponder what are the "blind spots" of this modeling approach, and discuss them, the cases where they may be more/less relevant, and potential workarounds. For example, suppose a training dataset where all graphs have a single node that is connected to most other nodes, whereas all other nodes' degree have an hard cap that is much lower than the block size. In the present algorithm, each diagonal block is generated independently, so there is no way to guarantee that exactly one such block will contain exactly one high-degree node. This is an example of a "blind spot", some structure that this model cannot capture. Other examples includes anything that involves at least 3 diagonal blocks or 2 non-diagonal ones.

Here is a variation that I think could eschew many of these limitations. We can view the problem as the autoregressive generation of blocks (diagonal or not) one-by-one, where generating each block demands to perform DDIM or DDPM. In this view, this paper chooses to generate all diagonal blocks first, then generate the off-diagonal ones conditional on the two diagonal blocks they each connect. However, other orders are possible, and the conditioning could *a priori* depend on all previously generated blocks using, e.g., GNNs. If the graphs are sparse, there is a regime where these GNNs can be kept in check while preserving the $C^2$ memory complexity.

**Other Strengths And Weaknesses:**

There are typos, inconsistencies and/or lack of clarity at critical points in the manuscript. I have already mentioned Algorithm 2's deficiencies elsewhere. Figure 2 uses $\mathbb{P}$ to represent the forward (noising) process, and $\mathbb{Q}$ for the backward (denoising) one, but the unnumbered equations in Section 2.1 use the opposite. The caption of Figure 2 does not go in enough details to help me understand the model (and it has another typo in the expression for $\mathbb{Q}$).

Further ablation studies are lacking, notably comparing random partitioning vs METIS. I don't currently understand the details as to how the off-diagonal blocks are obtained, but when I'll do I'll probably wonder what happens if this network is given as much capacity as the diagonal one (or the converse).

**Questions For Authors:**

My score of 2 indicates that, in it's current state, I would reject this manuscript, but I do see a possible path forward for me to recommend acceptance. This path depends on the answers to the following questions.

### Question 1

Do you disagree with, or wish to make any clarification about, any assessment I made in this review? If yes, could you please clarify your perspective?

### Question 2

Is an actual forward (noising) process involved for the off-diagonal adjacency matrices? Is the inference done in one-shot?

### Question 3

Can you clarify/fix Algorithm 2? Please be explicit, using $i$ and $j$ instead of $1$ and $2$ to clearly indicate how to proceed when there are more than two blocks. In other words, I should be able to read the "Computation Complexity" of Figure 1 from the loop structure. Please pay special attention to function signatures: if $s_\theta$ accepts 4 arguments and returns one output in Algorithm 1, it should accept 4 arguments and return one output in Algorithm 2.

You *may* do the same for Algorithm 1, especially if you notice some consistency/correctness issues while fixing Algorithm 2.

### Question 4

Can you provide all experimental details surrounding Figures 3 and 4? Please see my comments on Claims 3 and 4 above for details.

**Relation To Broader Scientific Literature:**

I've been out of date with the subfield for the last 2 years. The paper that I know should be cited, are cited. What is surprising is the absence of papers that I don't know: except for Aref & Mostajabdaveh (2024) cited in appendix, it looks like I didn't miss much in the last two years... That, or there are missing references!

**Theoretical Claims:**

I looked at all equations in the main text, but did not check any proofs in details. I gave some more attention to the algorithms, and found them wanting.

---

> ### Author Rebuttal · Authors · 2025-04-01
>
> Dear Reviewer,
>
> Thank you very much for your knowledgeable comments and insightful feedback. We greatly appreciate the time and effort you’ve invested in evaluating our work. Below, we address your concerns and questions. Please note that the responses may not follow the exact order of your original remarks, but we have ensured all points are addressed thoroughly.
>
> ## Algorithms for Training and Sampling
>
> We sincerely apologize for the confusion caused by the outdated pseudocode and associated inaccuracies in the description of our sampling procedure. Below, we clarify both the training and sampling processes, and we will revise the pseudocode accordingly in the updated manuscript.
>
> - **Training Process:** Our training approach follows a strategy similar to ClusterGCN [1]. At each iteration, we randomly sample two diagonal blocks and perform score learning using the diffusion model. The interaction between these two blocks (i.e., the off-diagonal component) is not subjected to the diffusion process but is instead inferred using a lightweight interaction network in a one-shot manner. As discussed in the paper, the motivation for this design is rooted in the fact that off-diagonal interactions tend to be sparse and easier to learn directly.
> - **Sampling Process:** Once the score networks and interaction networks are trained, graph generation are very flexible. First, one can generate the $k$ diagonal blocks using any preferred sampling method (e.g., DDPM, DDIM, or other variants). Then, the off-diagonal connections between each pair of blocks are generated using the interaction network.
>
> We regret the confusion and will ensure the corrected pseudocode and explanations are clearly reflected in the revision.
>
> ## Experimental Setup for Figures 3 and 4
>
> Thank you for pointing this out. Both Figures 3 and 4 use graphs generated from the cSBM model; however, the training graph sizes differ slightly: In Figure 3, training graphs have a size of 100. In Figure 4, training graphs have a size of 180, which matches the size used in our main experiments (e.g., Table 2).
>
> Additionally, the x-axis labels in Figure 4 should be interpreted as scaled by 100. We apologize for the oversight and will clarify this in the revised version of the manuscript.
>
> ## Limitations of the Model and Potential Extensions
>
> Thank you for your insightful critique regarding the model’s blind spots.
>
> We fully acknowledge that our approach—emphasizing assortative, block-wise dense structures—may not effectively capture certain types of graphs, particularly disassortative networks or structures like star graphs. These represent important limitations of the current modeling framework.
>
> That said, our structural prior is not arbitrary—it reflects common empirical patterns observed in many real-world networks, where assortativity and local structural cohesion are prevalent. In such domains, our assumptions are well-aligned with the underlying data distribution, contributing to the strong empirical results.
>
> We find your proposed variation—autoregressive generation of both diagonal and off-diagonal blocks, with conditioning on previously generated content using GNNs—particularly compelling and interesting! This approach could capture richer inter-block dependencies and remain scalable controlled memory use. This is an exciting direction for future research, and we are grateful for your thoughtful suggestion.
>
> ## Ablation on the Partition Algorithm
>
> Our choice of the METIS partitioning algorithm is motivated by our core modeling assumption: dense and homogeneous intra-block (diagonal) structure, with sparse inter-block (off-diagonal) connections. METIS is well-known for effectively partitioning graphs into balanced clusters while minimizing edge cuts—making it a natural fit for our setting.
>
> In early experiments, we did explore random partitioning, but found that it yielded significantly worse performance, likely due to poorly aligned block structures and noisier diffusion learning. We will try to include a more systematic study on this in the revision.
>
> Once again, we sincerely thank you for your constructive feedback. Your comments have helped us identify critical areas for clarification and potential extension, and have significantly improved the clarity, correctness, and quality of our manuscript.

---

> > ### Comment · Reviewer_KH7S · 2025-04-03
> >
> > Thank you for the clarifications. With the understanding that the camera ready will be adapted accordingly, I hereby increase my Overall Recommendation from 2 to 4.

---

> > > ### Author Response · Authors · 2025-04-04
> > >
> > > Again, thank you very much for taking the time to review our submission and for your thoughtful feedback. We sincerely appreciate your reconsideration and the decision to raise your scores. We will make sure to incorporate the points from our discussion in the revised version.

---

### Official Review · Reviewer_8djj · 2025-03-13

**Overall Recommendation:** 3

**Summary:**

This paper proposes a block graph representation on top of the diffusion model framework for graph generation. It claims to resolve the scalability and size generalization problems of existing models and comes with experiments on various benchmark datasets. The paper is generally well-written and easy to follow.

**Claims And Evidence:**

1. This paper claims to address the "scalability" concern of current graph generative models. To support this claim, it would be helpful to fully validate the approach with empirical experimental results. Could the authors compare the graph sizes in their benchmark datasets? For example, QM9 is relatively small in terms of average graph size (#nodes), while on the OGBN-products dataset, only the proposed method appears to train successfully without encountering an OOM error. Could the authors provide more details?

2. The paper claims to improve "size generalization" capability, demonstrated by the quantitative results in Fig. 4(a), where FID is used. However, since FID was originally proposed for image generation, it would be helpful if the authors could clarify what type of pre-trained models were used to compute FID in the context of graph evaluation. Providing these details would ensure a fair and rigorous comparison.
Also, what are the experimental settings for Fig. 4(a)?

**Essential References Not Discussed:**

More recent papers on graph diffusion models should be included as baselines for quantitative performance comparisons and technical discussions.

[1] Yan Q, Liang Z, Song Y, Liao R, Wang L. Swingnn: Rethinking permutation invariance in diffusion models for graph generation. arXiv preprint arXiv:2307.01646. 2023 Jul 4.

[2] Laabid N, Rissanen S, Heinonen M, Solin A, Garg V. Equivariant Denoisers Cannot Copy Graphs: Align Your Graph Diffusion Models. InThe Thirteenth International Conference on Learning Representations.

[3] Zhao L, Ding X, Akoglu L. Pard: Permutation-invariant autoregressive diffusion for graph generation. arXiv preprint arXiv:2402.03687. 2024 Feb 6.

**Experimental Designs Or Analyses:**

Already explained in the above comments.

**Methods And Evaluation Criteria:**

1. As mentioned above, some benchmark datasets may be too small to effectively validate the proposed method's ability to address the scalability issue. Please provide a clear comparison of their sizes.

2. Please explain the details of how FID is used for graph data.

3. Given that FCD and NSPDK (e.g., in [1,2]) are commonly used metrics for molecular generation tasks, could the authors clarify why these were not considered in the evaluation?

[1] Yan Q, Liang Z, Song Y, Liao R, Wang L. Swingnn: Rethinking permutation invariance in diffusion models for graph generation. arXiv preprint arXiv:2307.01646. 2023 Jul 4.

[2] Jo J, Lee S, Hwang SJ. Score-based generative modeling of graphs via the system of stochastic differential equations. InInternational conference on machine learning 2022 Jun 28 (pp. 10362-10383). PMLR.

**Other Comments Or Suggestions:**

N/A

**Other Strengths And Weaknesses:**

How is this paper different from previous work? Conceptually, it appears quite similar due to the decomposition process.

[1] Zhao L, Ding X, Akoglu L. Pard: Permutation-invariant autoregressive diffusion for graph generation. arXiv preprint arXiv:2402.03687. 2024 Feb 6.

**Questions For Authors:**

Already explained in the above comments.

**Relation To Broader Scientific Literature:**

I don’t have any critical comments.

**Theoretical Claims:**

The authors make several claims in Section 3.3: Theoretical Discussion, and below are my concerns.

1. First, it would be valuable to report the actual memory usage during both training and inference. On the same hardware, how does the model’s running speed compare to the baselines? While the big-O analysis in Table 1 is informative, there may be a gap between theoretical complexity analysis and practical deep neural network deployment. Additional details on real-world computational efficiency would strengthen the discussion.

2. The paper claims to have advantages in terms of "Distributed Training." However, I do not see any real experiments supporting the claim that "each computational node or device in a distributed system can handle smaller subproblems." While this may be conceptually true, the claim is fundamentally about practical implementation and must be validated by experiments to be considered valid.

---

> ### Author Rebuttal · Authors · 2025-04-01
>
> Dear Reviewer,
>
> Thank you very much for your detailed comments and thoughtful questions! They significantly help us improve the clarity and rigor of our manuscript. We appreciate the opportunity to address your concerns and questions below:
>
> ## Details on Graph Sizes in Datasets
>
> We apologize for omitting this information from the manuscript. The graph sizes used are publicly available, given the datasets utilized:
>
> 1. **Planar:** Graphs with 64 nodes.
> 2. **cSBM:** A synthetic graph dataset where graph size is configurable. For the main experiments (Table 2), we set the graph size to 180.
> 3. **QM9:** Graphs with sizes up to 9 nodes.
> 4. **OGBN-Arxiv:** Graphs up to 169,343 nodes.
> 5. **OGBN-Products:** Graphs up to 2,449,029 nodes, significantly larger than other datasets, which explains the "Out of Memory (OOM)" issues faced by most methods.
>
> We will include this explicitly in the revised manuscript.
>
> ## Clarification on FID Metric for Graph Evaluation
>
> We acknowledge your valid concern regarding the adaptation of FID (Frechet Inception Distance) from image to graph data. Following established practices in the literature ([1]), we adapted FID for graphs by utilizing a Graph Isomorphism Network (GIN) with random initialization as the embedding model. The embeddings were then used to compute FID scores following the standard procedure. We will clearly describe this approach in the updated manuscript.
>
> Regarding FCD and NSPDK, these metrics are specifically designed for molecular graphs, whereas our datasets primarily involve general graph structures. Thus, FID was chosen as a more universally applicable evaluation metric.
>
> ## Experimental Setup of Figure 4(a)
>
> For Figure 4(a), we used models trained on the cSBM dataset (graph size of 180) from Table 2, subsequently generating graphs of varying sizes. Please note the x-axis values represent graph size multiplied by 100. Figure 4 (a) shows that our method has the best size generalization performance among the tested methods.
>
> We apologize for any confusion and will clearly state this in the revised manuscript.
>
> ## Memory Usage and Computational Efficiency
>
> We appreciate your suggestion regarding memory usage. Alongside the theoretical analysis, we reported relative memory usage ratios from the experiments in Table 2's last column. We acknowledge discrepancies between theoretical predictions and practical performance, which stem from specific implementation details and hyperparameter selections. The revision will provide additional details, explicitly discussing runtime and implementation-dependent factors.
>
> ## Advantage for Distributed Training
>
> Due to technical constraints, we could not empirically verify advantages in distributed training scenarios. Therefore, we presented this as a theoretical discussion point rather than a main claim. This perspective will be clearly stated in the revised manuscript.
>
> ## Distinction from PARD
>
> Thank you for bringing PARD to our attention. While conceptually similar due to the principle on block structure, PARD represents a distinct class of graph generative models. It employs iterative component-based graph generation, aligning more closely with autoregressive models such as GraphRNN and EDGE. As such, PARD still require the entire graph in memory during generation, as noted in the original PARD paper ([2]). Our work specifically explores score-based diffusion generative models, thus positioning itself orthogonally to PARD.
>
> We appreciate the recommendation and will cite and discuss PARD clearly in the related work section of our revised manuscript.
>
> We greatly appreciate your insightful feedback and believe our revisions fully address your questions and concerns.
>
> [1] "On Evaluation Metrics for Graph Generative Models." *International Conference on Learning Representations (ICLR)*.
>
> [2] "PARD: Permutation-invariant Autoregressive Diffusion for Graph Generation," arXiv preprint arXiv:2402.03687, 2024.

---

### Decision · Program_Chairs · 2025-05-01

**Decision:**

Accept (poster)

**Comment:**

This paper presents SBGD, a diffusion-based graph generative model leveraging stochastic block structures to address key challenges in scalability and size generalization. The reviewers broadly appreciate the paper’s core idea of decomposing graphs into block representations to reduce memory overhead, enabling training on large-scale graphs. Empirical results support claims of competitive performance with improved efficiency.

Following the rebuttal, all reviewers either maintained or increased their scores, indicating that key concerns were satisfactorily addressed.